# Algorithmic progress in language models

**Anson Ho**[1*]  **Tamay Besiroglu**[1,2*]  **Ege Erdil**[1]  **David Owen**[1]

**Robi Rahman**[1]  **Zifan Carl Guo**[2]  **David Atkinson**[1,3]  **Neil Thompson**[2]

**Jaime Sevilla**[1]

## Abstract

We investigate the rate at which algorithms for pre-training language models have improved since the advent of deep learning. Using a dataset of over 200 language model evaluations on Wikitext and Penn Treebank spanning 2012-2023, we find that the compute required to reach a set performance threshold has halved approximately every 8 months, with a 90% confidence interval of around 2 to 22 months, substantially faster than hardware gains per Moore's Law. We estimate augmented scaling laws, which enable us to quantify algorithmic progress and determine the relative contributions of scaling models versus innovations in training algorithms. Despite the rapid pace of algorithmic progress and the development of new architectures such as the transformer, our analysis reveals that the increase in compute made an even larger contribution to overall performance improvements over this time period. Though limited by noisy benchmark data, our analysis quantifies the rapid progress in language modeling, shedding light on the relative contributions from compute and algorithms.

## 1  Introduction

The field of language modeling has seen rapid advances, with recent large language models (LLMs) demonstrating strong performance in domains such as programming [Li et al., 2022, Leblond et al., 2023], mathematics [Cobbe et al., 2021, Trinh et al., 2024], and standardized tests [OpenAI, 2023]. But how has this progress been possible?

One salient factor is the scaling of training compute based on neural scaling laws [Sevilla et al., 2022, Hoffmann et al., 2022, Kaplan et al., 2020], with state-of-the-art systems being trained for months on tens of thousands of GPUs. But this is only part of the story: another key factor has been algorithmic improvements, which result in more efficient use of resources such as compute and training data. These include changes in model architectures, optimization algorithms, and software frameworks.

This picture raises some important questions: How much of recent progress in language models has come from algorithmic improvements during pre-training, and how much has been from scaling up models and datasets? The answers to these questions have crucial implications for the future of AI progress, and have an important role in informing AI policy.

---

*Joint first authors. [1]Epoch. [2]MIT FutureTech, CSAIL, [3]Northeastern University. Email correspondence to `tamay@epochai.org`. You can find our code and data here: `https://github.com/epoch-research/lm-algorithmic-progress`.

38th Conference on Neural Information Processing Systems (NeurIPS 2024).

In this paper, we aim to answer these questions by following the approach first presented by Erdil and Besiroglu [2022] in computer vision.[2] To this end, we produce a dataset of over 200 language models evaluated on popular language modeling datasets, and use this to fit a statistical model that helps estimate the rate of algorithmic progress. Using our model, we further quantify the relative importance of algorithms and scaling training compute, shedding light on one of the most important drivers of AI progress.

## 1.1 Previous work

Thus far, there have been few works investigating algorithmic progress in machine learning. Notably, Hernandez and Brown [2020] re-implement popular open-source ImageNet models and find a $44\times$ decrease in the compute required to achieve the same performance as AlexNet. Karpathy [2022] reproduced the convolutional neural network of LeCun et al. [1989] using modern algorithmic innovations, achieving a 60% reduction in error rate. Dorner [2021] measures progress in the sample efficiency of deep reinforcement learning algorithms over time with doubling times ranging from 5 to 18 months. More recently, Erdil and Besiroglu [2022] estimate algorithmic progress based on fitting a statistical model inspired by neural scaling laws. They find that algorithms and hardware contribute roughly equally to performance, and the training compute needed to reach a certain level of performance halves every 9 months.

## 2 Methodology

### 2.1 Model definitions

We want to estimate the rate at which newer language models are able to achieve a certain level of performance more efficiently than older models. We do this by fitting a model that meets two key desiderata: (1) the model must be broadly consistent with previous work on neural scaling laws (e.g. Hoffmann et al. [2022]), and (2) the model should allow for a decomposition of the main contributors to increased performance, such as improvements in how efficiently data or free parameters in the model are used. In this sense, our core approach is similar to that in Erdil and Besiroglu [2022].

The starting point is the scaling law from Hoffmann et al. [2022], which relates the training loss $L$ of a dense transformer to its number of parameters $N$ and the training dataset size $D$:

$$L = E + \frac{A}{N^\alpha} + \frac{B}{D^\beta}, \tag{1}$$

where $L$ is per-token cross entropy loss on the dataset, and $E$, $A$, $B$, $\alpha$ and $\beta$ are constants. $E$ represents the 'irreducible loss' of the dataset, while the second and third terms, $\frac{A}{N^\alpha}$ and $\frac{B}{D^\beta}$, capture the errors that are due to the finiteness of the model or dataset, respectively.

Following Erdil and Besiroglu [2022] and Hernandez and Brown [2020], we quantify algorithmic progress in terms of reductions of the resources ($N$ and $D$) required to achieve the same level of performance over time. To measure this, we introduce the concepts of "effective data" $D_{\text{eff}}$ and "effective model size" $N_{\text{eff}}$ into the model:[3]

$$N_{\text{eff}} \equiv N \exp(\alpha'(Y - Y_0)), \text{ and } D_{\text{eff}} \equiv D \exp(\beta'(Y - Y_0)), \tag{2}$$

where $Y$ is the current year, $Y_0$ is some reference year[4], and $\alpha'$ and $\beta'$ characterize the rate of algorithmic progress for model size and dataset size, respectively. In other words, we assume that continued algorithmic progress results in an exponential increase in $D_{\text{eff}}$ and $N_{\text{eff}}$ over some time interval $Y - Y_0$, even with fixed $D$ and $N$. Plugging these into the original scaling law gives:

$$L = E + \frac{A}{N_{\text{eff}}^{\alpha_{\text{param}}}} + \frac{B}{D_{\text{eff}}^{\beta_{\text{data}}}} = E + \frac{A}{N^{\alpha_{\text{param}}}} e^{-\alpha_{\text{year}}(Y - Y_0)} + \frac{B}{D^{\beta_{\text{data}}}} e^{-\beta_{\text{year}}(Y - Y_0)}, \tag{3}$$

---

[2]Note that we focus on pre-training algorithmic progress, which is distinct from algorithmic progress in general. In particular, we do not consider "post-training enhancements" such as chain-of-thought prompting [Davidson et al., 2023].

[3]This is not an original idea—for example, Hernandez and Brown [2020] and Erdil and Besiroglu [2022] use the concept of "effective compute" to calculate doubling times for compute efficiency in computer vision, and Davidson [2023] incorporates a similar idea into an integrated economic model.

[4]Note that the "years" in our model do not need to be integers, i.e. "fractions of a year" are allowed and are determined based on the specific publication date of a model.

where $A$, $B$, $\alpha_{\text{param}}$, $\alpha_{\text{year}}$, $\beta_{\text{data}}$ and $\beta_{\text{year}}$ are constants. In relation to equation 2, we have that $\alpha' = \alpha_{\text{year}}/\alpha_{\text{param}}$ and $\beta' = \beta_{\text{year}}/\beta_{\text{data}}$. Algorithmic progress is thus captured as a constant exponential trend that multiplies with each of the two terms in the scaling law. In doing so, we are able to estimate the rate at which fewer 'resources' ($N$ and $D$) are required to achieve the same level of performance over time. Furthermore, given that the physical compute is approximately given by $C \approx 6ND$ [Hoffmann et al., 2022, Kaplan et al., 2020], we can similarly define an "effective compute" which is determined from the effective parameters and effective data.

## 2.2 Estimation approach

### 2.2.1 Model selection

We estimate variants of the augmented scaling law presented in equation (3) on our dataset of language model evaluations. We perform extensive cross-validation exercises to identify the variant of the model that fits the data best. The goal of this exercise is to consider different models that capture different effects (e.g. different scaling behavior across different model architectures, different forms of algorithmic progress, etc.).

Concretely, we consider dataset-specific coefficients $(A, B)$, rates of algorithmic progress (e.g. $\alpha_{\text{year}}, \beta_{\text{year}}$), different scaling coefficients for different architectures, regularization ($\alpha_{\text{param}}, \beta_{\text{data}}$), and more. The model variants we consider generally do not contain an irreducible loss term (i.e. $E = 0$) since this is poorly estimated on our data, and because it does not change our estimated doubling times in practice—we check the robustness of this change in appendix H. In total, we evaluate around 90 different model specifications through leave-one-out-cross validation and pick the models that perform best on relevant out-of-sample metrics, see Appendix J for more details. In the end, the model we select is model 7, where the coefficients $A$ and $B$ are benchmark specific, but estimates of algorithmic progress and scaling exponents (e.g. $\alpha$ and $\beta$) are not. This model achieves an $R^2$ of around 0.91 between predictions and held-out test data.

A further important consideration is the possibility of alternative forms of algorithmic progress. In particular, in section 2.1 we model algorithmic progress as causing exponential increases in an "effective" budget, e.g. of parameters. But one could also observe progress through changes in scaling exponents (i.e. $\alpha_{\text{param}}$ and $\beta_{\text{data}}$). There are *a priori* reasons to suspect that this might be the case—for instance, one notable innovation is due to a change in scaling laws such as those introduced in Kaplan et al. [2020] and Hoffmann et al. [2022]. Different model architectures, such as recurrent neural networks and transformers, are also known to have different scaling behaviors (see for instance Tay et al. [2022] and Droppo and Elibol [2021]).

We attempt to account for this possibility in the cross validation analysis. In particular, we introduce three models (models 13 to 15) which account for different kinds of scaling exponents, including the possibility of changing exponents over time. Our chosen main model (model 7) outperforms these models in cross validation, but these alternatives also perform similarly well, typically with an $R^2$ of between 0.88 and 0.91. This analysis is described in more detail in appendix J.

We also consider other factors that could potentially impact measured perplexity, and thereby measured rates of algorithmic progress. For example, different tokenization schemes during preprocessing have been found to improve WT103 perplexity in some instances [Radford et al., 2019], and training models for multiple epochs has been a common way of improving performance [Muennighoff et al., 2023]. We find that our core results are broadly the same while varying these degrees of freedom—we provide more details on these experiments in the appendices.[5]

### 2.2.2 Data

Our dataset contains over 400 language models evaluated on WikiText-103 (WT103), WikiText-2 (WT2), and Penn Treebank (PTB), about 60% of which we are able to use in our analysis. In particular, relevant information was retrieved from around 200 different papers, as well as evaluations of 25 models that we performed ourselves using the framework from Gao et al. [2021]. We then consider the subset of the data that contains the information necessary to fit our proposed model structure in equation 3: token-level test perplexity (which determines the cross-entropy loss), publication date,

---

[5]In particular, we consider tokenization in appendix E.2.2, epochs in appendix F, and context length in E.2.1.

number of model parameters, and training dataset size. This leaves us with around 231 models for analysis.

In some instances, multiple models are retrieved from the same paper, even if they constitute similar algorithmic innovations. This could pose problems around autocorrelation, which could result in underestimating the uncertainty in our individual parameter estimates. In the following main analysis, we therefore only include up to three models per paper, which results in approximately 50 more models being excluded. To verify the robustness of this approach, we also consider an alternative technique that directly accounts for autocorrelation in the analysis, which yields doubling time and confidence interval estimates that are consistent with our main results (see Appendix I).

## 3  Empirical results

### 3.1  Models require $2\times$ less compute roughly every eight months

How quickly are the algorithms underpinning language models improving? Our core approach is to back out doubling times based on fitting the augmented scaling law introduced in equation (8), and using the definitions of effective data, effective parameters, and effective compute we introduced in section 2.1. Here the effective data is given by $D_{\text{eff}} = D \exp\left[\frac{\beta_{\text{year}}}{\beta_{\text{data}}}(Y - Y_0)\right]$, so the doubling time for $D_{\text{eff}}$ is determined by the time $Y - Y_0$ where $D_{\text{eff}} = 2D$. Thus we have:

$$T_D = Y - Y_0 = \frac{\beta_{\text{data}}}{\beta_{\text{year}}} \ln 2. \tag{4}$$

The doubling times for parameter efficiency can be determined similarly, giving

$$T_N = \frac{\alpha_{\text{param}}}{\alpha_{\text{year}}} \ln 2, \tag{5}$$

which we can use to work out the doubling times for effective compute. In particular, since the total compute in FLOP, $C$, required during training is approximately $6ND$, the growth rates are related via $g_C = g_N + g_D$. Here $g_C$ is the growth rate in effective compute, $g_N$ is the growth rate in effective parameters, and $g_D$ is the growth rate in effective data. Since doubling times are inversely related to growth rates, we therefore have that

$$T_C = \left(\frac{1}{T_N} + \frac{1}{T_D}\right)^{-1}, \tag{6}$$

where $T_C$, $T_N$, and $T_D$ are the doubling times (due only to algorithmic progress in pre-training) for effective compute, effective parameters, and effective data respectively. Based on this approach, using our preferred model, we find that the median doubling time for effective compute is 6.1 months, with a 90% confidence interval of 3.3 to 11.3 months.

We further check the robustness of this result by looking at the predictions from different models. In particular, because we perform model selection using leave-one-out cross-validation, we can compare the predictions of our preferred model with the predictions from other models we considered.[6] Concatenating the doubling time estimates from all the models in Figure 1b, we find a median doubling time of 7.5 months [95% CI: 1.7 to 22.5 months], which is consistent with our preferred model.[7] This estimate falls within the range of confidence intervals of the estimated rates of algorithmic progress in computer vision [Erdil and Besiroglu, 2022], and sample efficiency improvements in reinforcement learning [Dorner, 2021].

While the structure of our model is not amenable to analyzing fine-grained speedups or slowdowns in the rate of algorithmic improvements, we can nevertheless test the possibility of a one-time increase or

---

[6]Note that our preferred model is model 7, whereas the model that performs best in cross validation is model 10. We opt for model 7 given that it performs essentially as well in cross validation (MSE test loss of $\sim$0.048 for model 7 compared to $\sim$0.042 for model 10) but uses two fewer parameters. In addition, model 7 can be used to back out a single rate of algorithmic progress, rather than dataset-specific rates, which makes the results easier to interpret. More details about the models and their performance can be found in appendix J.

[7]In both the preferred model and aggregated estimates, the lower range of the confidence interval implies extremely fast doubling times. While we believe there is reason to be sceptical of these estimates, we provide the estimates here for transparency. We elaborate on this concern in Section **??**.

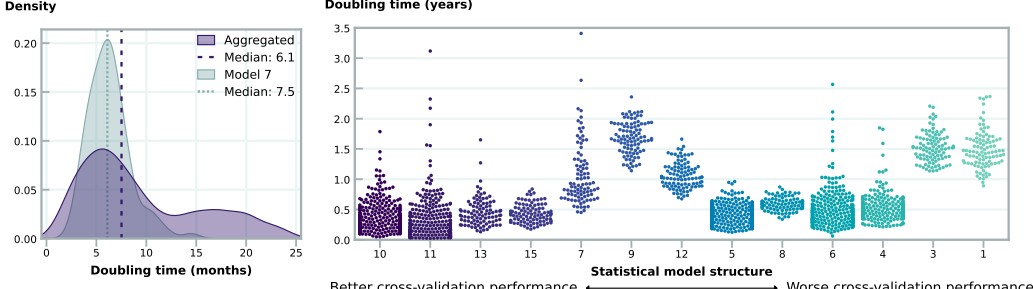

(a) **Core estimates.** Doubling times from our preferred model, and aggregate across model specifications.

(b) **Robustness across model specification.** Swarm plots showing model estimates of the rate of algorithmic progress across distinct model structures. For each model, we choose the regularization strength $\delta$ that performs best in leave-one-out cross validation.

| Degree of Freedom | 1 | 2 | 3 | 4 | 5 | 6 | 7 | 8 | 9 | 10 | 11 | 12 | 13 | 14 | 15 |
|---|---|---|---|---|---|---|---|---|---|---|---|---|---|---|---|
| Progress in Efficiency Along $N$ | ✓ | × | ✓ | ✓ | ✓ | ✓ | ✓ | × | ✓ | ✓ | ✓ | ✓ | ✓ | ✓$^T$ | ✓$^T$ |
| Progress in Efficiency Along $D$ | ✓ | ✓ | × | ✓ | ✓ | ✓ | ✓ | ✓ | × | ✓ | ✓ | ✓ | ✓ | ✓$^T$ | ✓$^T$ |
| Dataset Specific Exponents | × | × | × | ✓ | ✓ | ✓ | × | × | × | ✓ | ✓ | × | × | × | × |
| Dataset Specific Constants | × | × | × | × | × | × | ✓ | ✓ | ✓ | ✓ | ✓ | × | × | × | × |

(c) **Summary of all model structures** and the degrees of freedom included. Efficiency gains are captured by exponential decrease in the relevant error terms, except models indicated by $^T$, which have time-varying exponents. For a full specification, see Tables 11 and 12.

Figure 1: Estimates of algorithmic progress of models selected by cross validation. Figure 3a shows aggregated estimates over doubling times, and Figure 3b illustrates via swarm plots sorted from left to right in order of decreasing cross validation performance (increasing MSE test loss). Note that model 14 is omitted from Figure 3b —we elaborate on our reasoning in appendix J.2.

decrease in growth rates over the full time period. To this end, we consider a variant of our preferred model (model 7) where a dummy variable is introduced—this is equal to 0 for any model that is published before the start of a certain year, and 1 otherwise. This allows us to consider doubling times before and after a certain year cutoff (e.g. 2017), and we perform this analysis for several such cutoffs.

The result is shown in Figure 2. Here we see that the difference in estimated doubling time before and after the start of 2017 is very pronounced, however this is not the case for other choices of the cutoff year. In each year the median doubling time is faster after the start of the cutoff year, but usually only marginally so. Overall, this does not provide strong evidence of a drastic speedup in algorithmic progress. This does not rule out the possibility of weaker effect sizes, since our approach is statistically under-powered.

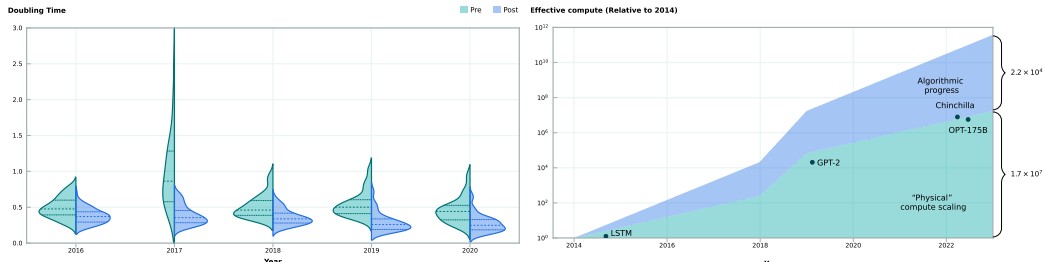

Figure 2: **Left**: Comparison of estimated doubling times for effective compute from algorithmic progress, before and after set cutoff years from 2016-2020. Shorter doubling times in the "post" period relative to "pre" indicate an acceleration in the rate of algorithmic progress after that cutoff year. Longer doubling times indicate a deceleration. **Right**: A stylized illustration of the relative contribution of compute scaling and algorithmic progress to effective compute. The physical compute contribution is estimated from the doubling times in Sevilla et al. [2022], and the algorithmic progress contribution is based on the aggregated doubling time estimate across model specifications (see section 3.1). We further plot the physical training compute values for several notable models (e.g. GPT-2) in their publication years.

### 3.2 Most recent performance gains in next-token prediction have been from compute-scaling

Naively extrapolating our estimated doubling times suggests that, between 2014 and 2023, pre-training algorithmic progress has enabled performance to improve as much as it would have with around 22,000× more compute.[8] At the same time, Sevilla et al. [2022] find that physical compute budgets have doubled roughly every 6 months since the start of deep learning, including in language models. This suggests that physical compute has instead grown by a factor of around one-million-fold. This paints a stylized picture where "effective compute" expanded by about 22-billion-fold since 2014, with slightly under two-thirds of the scaling being due to increased use of actual, physical computing resources.

There are reasons to be cautious about this naive extrapolation. For one, we do not directly observe gains of $22,000\times$ (or even $10,000\times$) anywhere in our dataset. However, given that it is unlikely that early researchers trained language models on very large quantities of compute, it is therefore improbable that we observe such large declines over the analyzed time period. Nevertheless, the lack of such observations still raises questions about the reliability of extrapolating these trends between long multi-year periods.

One specific reason for caution is that the extrapolation neglects the scale-dependence of algorithmic innovations. It is likely that some algorithmic innovations will become obsolete over time as models are trained at larger scales of compute—e.g. the effectiveness of specific tokenizers or hyperparameter settings may diminish, making them less useful for future, larger models. Conversely, recent innovations might fail to produce large or any benefits when implemented at much smaller scales than models today. For example, the gains from scaling laws are related to the scale of compute used (see Appendix B), and older architectures, such as the LSTM and convolutional network, can exhibit higher efficiency at small scales relative to the transformer [Droppo and Elibol, 2021, Karpathy, 2022].

While a naive extrapolation of doubling times predicts substantial reductions in compute requirements, our work does not provide compelling evidence that we can currently or in the future train extremely small models to achieve the performance of much larger ones by applying the full suite of modern innovations. The scale-dependence of algorithmic improvements and the lack of direct observations of such large efficiency gains in our dataset suggest that further research and more comprehensive data are needed to validate these extrapolations.

Besides doubling times, we can also decompose the relative contributions from algorithms and compute scaling by evaluating our estimated models directly. We approach this using a Shapley value analysis, and the results weakly support the stylized picture above that compute scaling has been more important for explaining performance improvements than algorithmic progress since 2014.

The findings indicate that the relative contribution of algorithmic progress to performance improvements has diminished over time, at least within the dataset of models that have historically been close to the state-of-the-art. This observation aligns with the stylized representation in Figure 2 and the findings of Erdil and Besiroglu [2022] for computer vision, where compute scaling has shown increasing importance over time.

One explanation for the diminishing relative contribution of algorithmic progress is that investments in expanding physical compute have increased substantially, outpacing the rate of algorithmic improvements. This framing aligns with the increased emphasis on scaling large language models over the last few years, particularly since the introduction of GPT-2 in 2019 [Radford et al., 2019], relative to fundamental algorithmic or architectural changes.[9] Figure 2 illustrates a stylized version of this perspective, depicting a sharp increase in physical compute scaling around 2018-2019, followed by a return to previous compute scaling growth rates.

There are other potential explanations – for example, it is possible that the transformer architecture was a pivotal innovation (see section 3.3), and subsequent algorithmic advances have been less

---

[8]We consider 2014 since this is publication year of the earliest model in our dataset for which the training compute is known.

[9]We can provide further support for this interpretation by considering the average growth in compute between pairs of systems in Table 1. This turns out to be higher for later pairs of systems that we consider: e.g. between the Transformer and GPT-3 there was an average annual growth rate of 9%, compared to an average growth rate of 2% between the 2012 RNN and GPT-2.

|  | Parameter scaling | Data scaling | Data efficiency |
|---|---|---|---|
| RNN (2012) → LSTM (2016) | 16.9% | 40.4% | 43.8% |
| RNN (2012) → Transformer (2018) | 48.1% | 20.6% | 32.1% |
| RNN (2012) → GPT-2 (2019) | 47.7% | 20.1% | 32.9% |
| RNN (2012) → GPT-3 (2021) | 50.2% | 26.0% | 24.4% |
| RNN (2012) → Gopher (2021) | 54.8% | 24.0% | 21.7% |
| LSTM (2016) → Transformer (2018) | 82.6% | 0.0% | 17.9% |
| LSTM (2016) → GPT-2 (2019) | 72.1% | 16.1% | 12.1% |
| LSTM (2016) → GPT-3 (2021) | 69.9% | 20.0% | 10.3% |
| LSTM (2016) → Gopher (2021) | 68.8% | 17.6% | 13.9% |
| Transformer (2018) → GPT-2 (2019) | 56.8% | 38.2% | 5.0% |
| Transformer (2018) → GPT-3 (2021) | 63.1% | 29.7% | 7.4% |
| Transformer (2018) → Gopher (2021) | 61.9% | 25.4% | 12.9% |

Table 1: Attribution of progress to pre-training algorithmic progress and compute scaling between model pairs based on Shapley decomposition in linear space. Numbers may not all add up to 100% due to rounding. These Shapley values are based on point estimates from our preferred model and as such are meant for illustrative purposes only. We omit parameter efficiency improvements from the table since these are almost always 0% and not very informative. The Transformer here is by Baevski and Auli [2018] (the earliest decoder-only transformer we have in our dataset), who modify the original transformer architecture by Vaswani et al. [2017] to be decoder-only.

significant in comparison. Alternatively, this observation could also be explained by a secular decline in the rate of algorithmic innovation. However, we find these two explanations less compelling than the results of Figure 2, where the rate of algorithmic progress does not clearly decrease after the release of the transformer (e.g. with a 2018 cutoff). If anything, the rate *increases* slightly, contrary to what both of these explanations predict.

## 3.3 The significance of the transformer architecture

Since its introduction in 2017 [Vaswani et al., 2017], the transformer architecture has become the dominant algorithmic architecture in language modeling, forming the base of multiple notable systems. We attempt to quantify the its contribution in terms of the "compute-equivalent gain" over other architectures in our dataset (LSTMs, RNNs, state space models, among others). This is akin to the approach outlined in Davidson et al. [2023]—in this context, the compute-equivalent gain is the amount by which training compute must be scaled to improve benchmark performance as the same amount as the introduction of the transformer. For example, Hernandez and Brown [2020] find that a transformer (2017) achieves the same performance as a Seq2Seq (2014) model on the WMT-14-EN-FR benchmark, with $61\times$ less compute.

To capture the improvement represented by the transformer, we modify our core model as follows:

$$L = \begin{cases} \sigma(\gamma_T) \left( \frac{A}{N_{\text{eff}}^{\alpha_{\text{year}}}} + \frac{B}{D_{\text{eff}}^{\beta_{\text{data}}}} \right), & \text{if transformer,} \\ \frac{A}{N_{\text{eff}}^{\alpha_{\text{year}}}} + \frac{B}{D_{\text{eff}}^{\beta_{\text{data}}}}, & \text{otherwise.} \end{cases} \quad (7)$$

where $\sigma : \mathbb{R} \to (0, 1)$ is the sigmoid function, given by $\sigma(x) = 1/(1 + e^{-x})$. $\gamma_T$ is a constant and all other terms have the same meaning as in the original model.[10] The key intuition is that the transformer could enable us to use compute (or perhaps data) more efficiently than the architectures that precede it.

After preprocessing, our dataset contains 103 transformer models, and 127 non-transformer models, largely consisting of recurrent networks such as the LSTM. Fitting the model on this data reveals that the transformer architecture typically lowers reducible loss proportionally by 5.4% [90% CI: 3.8%, 6.9%].

We can calculate its contribution in terms of "compute-equivalent gains" numerically: we first calculate the predicted loss for a transformer with some $N$ and $D$, and the predicted loss for a

---

[10]The sigmoid is introduced to make it easier to fit the model by improving optimizer stability.

non-transformer with the same inputs. We then determine reduction in $N$ and $D$ to match this difference in loss. Compute is then approximated as usual, as $C \approx 6ND$. In short, if an innovation halves the compute needed to achieve a specific loss, then that innovation has a compute-equivalent gain of 2.

Based on 100 bootstraps, we obtain a median estimate of $9.6\times$ [90% CI: $4.3\times$, $34.5\times$] for the transformer's compute-equivalent gain.[11] This substantial gain indicates that the efficiency offered by the transformer architecture is equivalent to around $\log(9.6)/\log(2e4) \approx 23\%$ of the total gains from algorithms in the past nine years, or nearly two years of algorithmic progress in the field.[12] Moreover, this could understate the gains if the transformer architecture also provides a convenient vehicle through which to productively channel compute, thereby facilitating some of the gains through the scaling of compute that have likely dominated the overall gains we have seen recently.

One caveat here is that the measured significance of the transformer may depend on how it is evaluated. For example, transformers may be better adapted to long contexts than recurrent networks, and evaluations using longer contexts (e.g. >1000 tokens) may suggest a larger improvement from transformers than evaluations using shorter contexts [Kaplan et al., 2020]. We have not explicitly controlled for context length here, and we discuss the potential impact of this assumption in more detail in appendix E.2.1.

## 4 Discussion and conclusion

### 4.1 Summary of our findings

This paper presents a comprehensive empirical analysis of algorithmic progress in language model pre-training from 2012 to 2023. By curating a dataset of over 200 language model evaluations on WikiText and Penn Treebank benchmarks, we quantify the relative contributions of compute scaling and algorithmic efficiency improvements to the overall performance gains. Our key findings are as follows:

First, we estimate that the compute required to reach a set language modeling performance level has halved every 7-8 months on average since 2012. This supports the common intuition that language modeling is an unusually rapidly-advancing field of computer science.

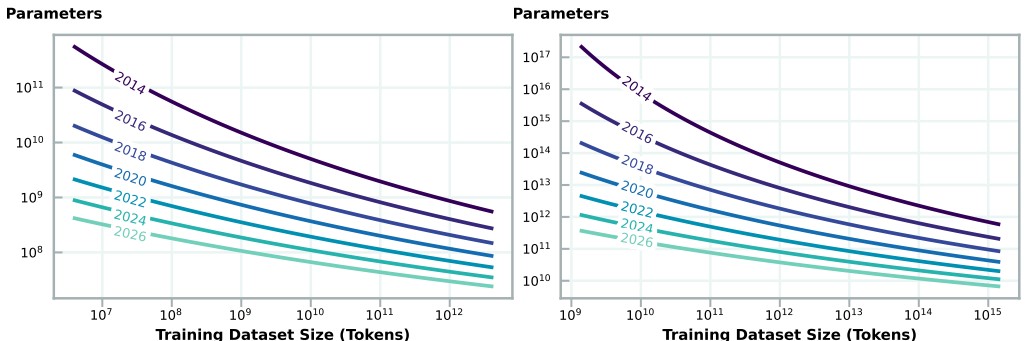

Predicted requirements for GPT-2 performance    Predicted requirements for Chinchilla performance

Figure 3: Pareto frontiers for GPT-2 [Radford et al., 2019] and Chinchilla [Hoffmann et al., 2022] level performance on WT103. We truncate the frontiers to a factor of 1e3 greater or smaller than the existing training dataset size and parameter size of the actual model since extrapolating further out would not be reliable.

Second, our work reveals that the majority of recent advancements in language modeling stem more from scaling models and datasets than from pre-training algorithmic innovations. A Shapley

---

[11]This assumes compute budgets of frontier models today, at $10^{25}$ FLOP. At lower compute budgets, such as $10^{22}$ FLOP, the gain is still substantial at $8.4\times$ [90% CI: $4.2\times$, $19.5\times$].

[12]Given the magnitude of this contribution, we also attempted to check the rate of algorithmic progress while subsetting our data to non-transformers only. However, this roughly halves the data available for fitting, and our resulting estimates are unfortunately extremely noisy: the estimated doubling times have a 90% confidence interval of $-11.7$ to $8.0$ months.

value-based analysis suggests that 60-95% of the performance gains stem from compute scaling, while algorithms contribute only 5-40%.

Third, the introduction of the transformer architecture in 2017 was a major algorithmic advance, representing between 3x and 46x in compute-equivalent gain, which accounts for more than 10% of the algorithmic innovation in pre-trained language models in the past decade. This highlights the significance of the transformer as a key architectural breakthrough in the field.

## 4.2 Limitations

While our analysis is an advance in quantifying algorithmic progress, several limitations reduce the precision of and temper our confidence in our estimates:

- **Lack of estimates of gains from specific innovations**. Our model is specified to quantify algorithmic progress over relatively large time periods (e.g. over several years). However, it is unable to give reliable fine-grained information, such as progress over shorter time scales, or the significance of specific innovations. Experimental work is better suited to estimating efficiency gains for specific algorithmic innovations.

- **Limited availability of quality data, resulting in noisy or unrealistic estimates of progress**. The approach we use in our analysis relies heavily on having many data samples across many years. This proved to be very challenging for a number of reasons—e.g. models are not always evaluated on the same benchmark, data is relatively sparse prior to 2017, and papers may not report relevant information such as parameter counts. Among other reasons this can result in our estimates being very noisy, yielding wide confidence intervals over doubling times. In addition, algorithmic improvements and scaling have historically been introduced concurrently, and this correlation between the two in our dataset can make it hard to disentangle their relative contributions to overall effective compute growth.

- **Inconsistencies in model training and evaluations**. Inconsistencies in evaluations are well-known. While we have excluded non-standard evaluations from our dataset, our dataset spans models with different tokenization schemes, text preprocessing, stride lengths, and other details. This introduces noise and potential bias in our estimates of algorithmic progress, as researchers might have adopted more favorable evaluation schemes over time. However, our estimated reductions in perplexity from algorithmic improvements are large; likely larger than can be accounted for by changes in evaluation procedures. We expand on these points in Appendix E.2.3.

- **Inability to distinguish between data quality and efficiency in data use**. Reduction in data requirements could be due to both improved data quality and improved algorithms, but our model is not equipped to distinguish between these effects. Understanding the relative contributions of each could be a subject of future research.

- **Reliance on the Chinchilla scaling law**. The scaling law from which our model is derived applies to dense transformers following a GPT-3 architecture [Hoffmann et al., 2022, Rae et al., 2021]. However, we use this scaling law to model algorithmic improvements in different transformer architectures, recurrent neural networks, etc. Future algorithms might also follow different scaling laws (e.g. GPT-4 is rumored to be a mixture of experts). However, we believe it is likely that our core results should still hold: for one, neural scaling is not a phenomenon restricted to transformers (e.g. it is known to happen in RNNs as well, see Kaplan et al. [2020]). We find that a wide range of statistical model structures provide consistent estimates, and that alternative methods of estimating pre-training algorithmic progress also give similar results (see e.g. appendix A), so it is probable that our core results are robust to the use of the scaling law from Hoffmann et al. [2022].

- **Limited insight about future progress**. While the results from this paper could be used to inform one about future progress in language modeling, our paper focuses on historical improvements. Future rates of progress could be slower (e.g. if one thinks that historical progress consisted of picking "low hanging-fruit"), but they could potentially also be faster (e.g. due to increased research interest and investment). Expectations about future progress need to account for factors such as these, which we do not discuss in depth for the most part.

## 4.3 Conclusion

Using a dataset of over 200 language model evaluations spanning 2012-2023 evaluated on Wikitext and Penn Treebank, we find that the compute required to reach a fixed performance threshold has halved approximately every 8 months. This is much faster than the rate associated with Moore's law and many other domains of computing. While algorithmic innovations have occurred rapidly, compute scaling has expanded by over a million-fold in this same period, exceeding the gains from algorithms and constituting the predominant source of performance improvements in recent years.

Overall, our work provides a quantitative estimate of the rapid pace of progress in language modeling. It also reveals the dominant role of scale rather than algorithms for recent gains. Future work could benefit from extending this analysis to additional, specific benchmarks and more closely examining the impact of data quality improvements and the gains from additional specific innovations. Despite its limitations, this research demonstrates the valuable insights that can be gained from a detailed statistical analysis of extensive datasets of machine learning results. By identifying the main drivers of performance improvements, this work lays the groundwork for further exploration and understanding of these trends in the field.

## Acknowledgments

We thank Tom Davidson, Pablo Villalobos, Josh You, Lukas Finnveden, Eli Lifland, nostalgebraist, David Schneider-Joseph, Danny Hernandez, Alyssa Vance, Yafah Edelman, Ben Edelman, Matthew Barnett, Ben Cottier, Keith Wynroe, Markus Anderljung, Carl Shulman, Marius Hobbhahn and Nikola Jurković for their feedback. We thank Eduardo Roldán and Robert Sandler for helping design and implement graphs.

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

# A    Observing improvements in the data

Besides the statistical model that we presented in section 2.1, we can also attempt to obtain doubling time estimates more directly. For example, we can look at LLMs that achieve close to Megatron-LM's or GPT-2's level of performance over time, and see how much less compute is used. Doing so reveals that we need between 5-fold and 100-fold less compute per year in 2023 to achieve the same performance achieved is between 2019 and 2023, amounting to a halving time of between 11 and 17 months. This is within the 90% confidence interval of our aggregate doubling time estimates, from 1.7 to 22.5 months.

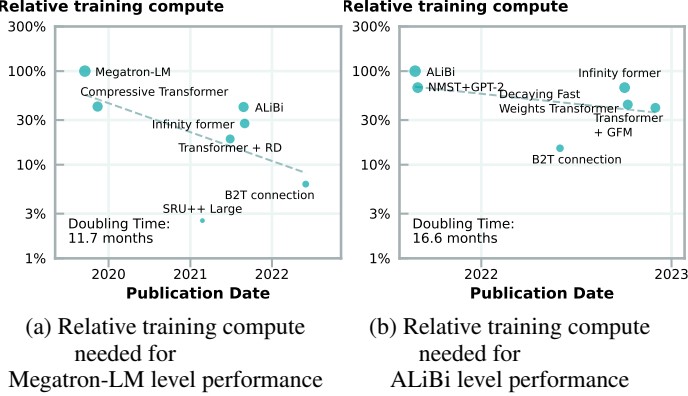

(a) Relative training compute needed for Megatron-LM level performance

(b) Relative training compute needed for ALiBi level performance

Figure 4: Relative compute (relative to baseline model) used to train models that achieve the same evaluated perplexity as Megatron-LM and ALiBi respectively. Doubling times of effective compute are 11.7 and 16.6 months using least squares regression for Megatron-LM (cross-entropy range 2.87-3.06) and ALiBi (cross-entropy range 1.18-1.34), respectively. Circles are proportional to the compute used during training.

This approach provides some insight but it has its issues, which is why we do not rely on it. For instance, it depends on the particular choice of reference performance. Moreover, this approach requires identifying models with similar or better performance at later dates, where the training compute is known. However, the data for the latter is relatively limited. We thus opt for using an arguably more principled approach with our core model presented in section 2.1 for our main results.

# B    The gains from better scaling laws

We estimated the compute savings afforded by the Chinchilla scaling law, as proposed by Hoffmann et al. [2022], in contrast to the previously dominant understanding based on the work of Kaplan et al. [2020]. First, we defined loss functions $L(N, D)$ for both the Kaplan and Chinchilla scaling laws. Following this, we minimized these loss functions across variables $D$ and $N$, considering different levels of compute budget. For each specified budget, we then calculated the amount of compute required under the Chinchilla scaling law to achieve a loss equivalent to the minimum loss obtained under the Kaplan scaling law. The Compute-Equivalent Gain (CEG) was subsequently determined as the ratio of the original compute budget to the compute required by the Chinchilla scaling to match the Kaplan loss.

We find that the compute equivalent multiplier from the Chinchilla scaling laws for dense autoregressive transformer models is between 1.25-fold (for GPT-2 scale models) and 1.6-fold (for PaLM-scale models Chowdhery et al. [2023]).[13]

---

[13]We use PaLM as a reference rather than larger more recent models such as GPT-4 because it was unlikely that GPT-4 would have been trained without an improvement in our understanding of scaling laws, whereas PaLM was likely trained prior to the development of updated scaling laws.

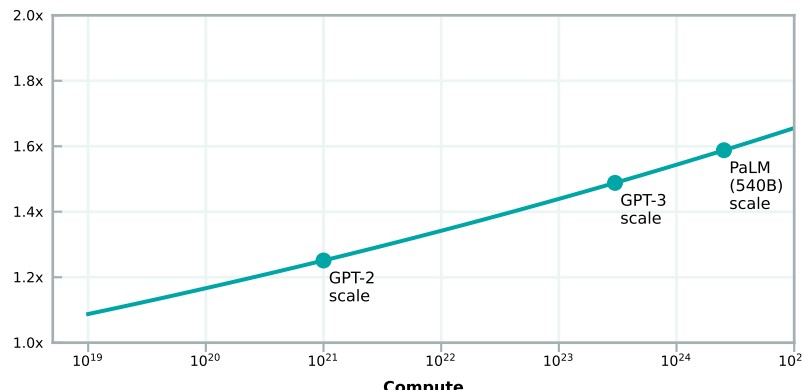

Figure 5: Compute equivalent multiplier from optimal scaling from switching from Kaplan et al. [2020] to Chinchilla (Hoffmann et al. [2022]) scaling laws as a function of training compute for dense autoregressive transformer models. Note that GPT-3 and PaLM (540B) use around 1.7 and 1.44 tokens/parameter respectively, close to what the Kaplan scaling laws recommend, suggesting that Kaplan-scaling was close to what was practiced at the time.

## C   Core model parameter estimates

The core model that we use was chosen based on leave-one-out cross validation, and is defined similarly to equation 3 but with a few modifications. The most important change is that $A$ and $B$ are estimated separately for each benchmark, whereas all other parameters are benchmark-agnostic. In order to help with model fitting, we normalize $N$ and $D$ to some minimum $N_0$ and $D_0$ values in our dataset, and reparameterize $A$ and $B$ as exponentials. In full, our model is

$$L = \exp[\alpha'_{\text{const}} - \alpha_{\text{year}}(Y - Y_0) - \alpha_{\text{param}} \log(N/N_0)] + \exp[\beta'_{\text{const}} - \beta_{\text{year}}(Y - Y_0) - \beta_{\text{data}} \log(D/D_0)], \tag{8}$$

To estimate these in benchmark-specific fashion, we introduce dummy variables $x_{\text{WT2}}$ and $x_{\text{PTB}}$ for WT2 and PTB respectively. We then complete the model definition as follows:

$$\alpha'_{\text{const}} = \alpha_{\text{const}} + \alpha_{\text{const,PTB}} x_{\text{PTB}} + \alpha_{\text{const,WT2}} x_{\text{WT2}},$$
$$\beta'_{\text{const}} = \beta_{\text{const}} + \beta_{\text{const,PTB}} x_{\text{PTB}} + \beta_{\text{const,WT2}} x_{\text{WT2}}.$$

Our parameter estimates are summarized in Table 2.

One observation about the parameter estimates in Table 2 is that the confidence intervals for $\alpha_{\text{year}}$ and $\beta_{\text{year}}$ are not statistically significant at the 5% significance level, while $\alpha_{\text{param}}$ and $\beta_{\text{data}}$ are. As mentioned in section 3.2, the result is that the model fails to obtain statistically significant estimates of effective parameter and effective data doubling times. However, when we use these estimates to determine effective compute doubling times, we obtain statistically significant estimates. The reason for this is illustrated in Figure 6—the estimates for $\alpha_{\text{year}}$ and $\beta_{\text{year}}$ are clearly negatively correlated. In particular, when $\alpha_{\text{year}}$ is positive, $\beta_{\text{year}}$ is negative and vice versa, such that the overall estimated effective compute doubling time is always positive.

### C.1   Comparing our estimates to earlier work

Given that our core model is similar to previously proposed language model scaling laws, we can compare our estimates to see how well they correspond to prior work. In particular, the estimates for $\alpha_{\text{param}}$ and $\beta_{\text{data}}$ in Table 2 suggest that cross entropy loss scales roughly as $C^{-1/20}$, where $C$ is training compute. In comparison, Kaplan et al. [2020] find a scaling exponent of around -0.048, and Hoffmann et al. [2022] estimate values of around -0.3. Given that our model is constructed based on the scaling law in Hoffmann et al. [2022], we might *a priori* have expected our estimated to match those more closely—so what explains the difference?

One way to understand this discrepancy is to consider the scaling laws on the same plot, shown in Figure 6. Here we observe that the scaling laws strongly diverge for compute values below around

|  | Estimate | 90% CI |
|---|---|---|
| $\alpha_{\text{const}}$ | 0.903 (0.232) | 0.647, 1.300 |
| $\alpha_{\text{const,PTB}}$ | −0.000 (0.139) | −0.264, 0.206 |
| $\alpha_{\text{const,WT2}}$ | 0.000 (0.114) | −0.213, 0.164 |
| $\alpha_{\text{year}}$ | −0.001 (0.021) | −0.032, 0.017 |
| $\alpha_{\text{param}}$ | 0.083 (0.017) | 0.058, 0.102 |
| $\beta_{\text{const}}$ | 0.791 (0.240) | 0.197, 1.057 |
| $\beta_{\text{const,PTB}}$ | 0.190 (0.201) | −0.010, 0.432 |
| $\beta_{\text{const,WT2}}$ | 0.163 (0.162) | −0.001, 0.425 |
| $\beta_{\text{year}}$ | 0.038 (0.022) | 0.017, 0.082 |
| $\beta_{\text{data}}$ | 0.030 (0.010) | 0.020, 0.050 |

Table 2: Parameter estimates from the model described in equation 8, rounded to 3 decimal places. We report 90% confidence intervals for all of the parameter estimates by bootstrapping 100 iterations.

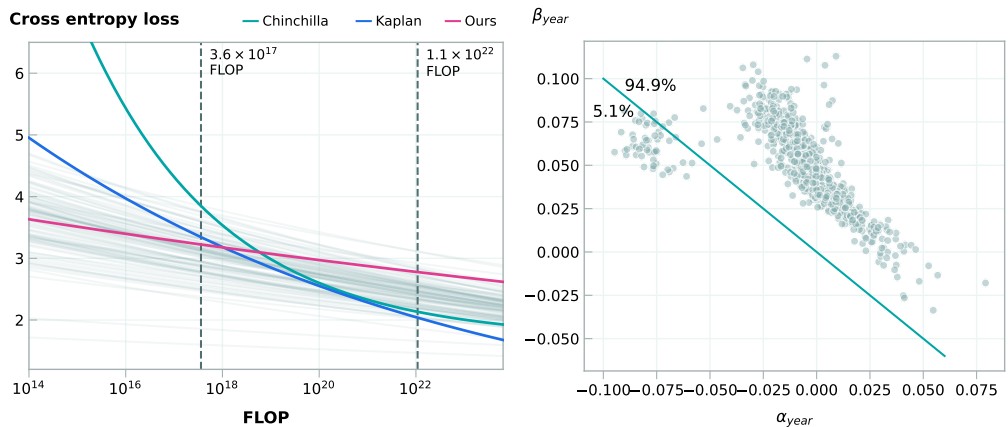

Figure 6: (a) (Left) Comparison of scaling law predictions from our preferred model and previous work, specifically Kaplan et al. [2020] and Hoffmann et al. [2022]. The grey lines represent scaling laws based on bootstraps of our proposed model. The two vertical dotted lines indicated the 10th to 90th percentile range of training compute values in our dataset. (b) (Right) Estimated values of $\alpha_{\text{year}}$ and $\beta_{\text{year}}$ from 1000 bootstraps. 94.9% of the bootstrapped estimates lie to the right of the line $\alpha_{\text{year}} + \beta_{\text{year}} = 0$.

$10^{18}$ to $10^{19}$ FLOP (and the same is true for values greater than $10^{22}$ FLOP). However, between these two regimes the scaling laws appear much more similar in slope.

This observation suggests the possibility that the discrepancy in estimated scaling exponents is due to the range of fitted data. Indeed, around 80% of our models with known training compute estimates lie between $\sim 4 \times 10^{17}$ FLOP and $10^{22}$ FLOP. This suggests that a large fraction of our data lies within the regime where it is hard for our model to distinguish between the exponents from Hoffmann et al. [2022] and Kaplan et al. [2020].

Another possible explanation for this discrepancy that we considered is that it is due to the omission of an irreducible loss term in our core model, resulting in an omitted variable bias. However we do not put much weight on this explanation for our fits—in our robustness check using models with an irreducible loss term (see section H), we obtain very similar scaling exponents to those obtained in our core model.

# D  Significance of the transformer architecture

Similarly to the doubling times for effective compute, we consider the predicted Compute-Equivalent Gains by applying the same modification in Equation 7 to different model specifications. Most models yield estimates within a similar ballpark as our core model (model 7 with $\delta = 0.0025$), with some models yielding relatively noisy estimates. That said, three models predict a notably larger efficiency contribution from the transformer, and this suggests that there is plausibly a fairly large degree of cross-model uncertainty present. These results are shown in Figure 7.

|  | Estimate | 90% CI |
|---|---|---|
| $\alpha_{\text{const}}$ | 0.358 (0.442) | $0.000, 1.257$ |
| $\alpha_{\text{const,PTB}}$ | 0.030 (0.148) | $-0.234, 0.151$ |
| $\alpha_{\text{const,WT2}}$ | 0.000 (0.095) | $-0.184, 0.133$ |
| $\alpha_{\text{year}}$ | $-0.048$ (0.031) | $-0.077, 0.018$ |
| $\alpha_{\text{param}}$ | 0.117 (0.032) | $0.052, 0.159$ |
| $\beta_{\text{const}}$ | 1.184 (0.381) | $0.207, 1.330$ |
| $\beta_{\text{const,PTB}}$ | 0.108 (0.161) | $-0.000, 0.299$ |
| $\beta_{\text{const,WT2}}$ | 0.101 (0.142) | $-0.000, 0.481$ |
| $\beta_{\text{year}}$ | 0.052 (0.021) | $0.017, 0.085$ |
| $\beta_{\text{data}}$ | 0.018 (0.014) | $0.012, 0.052$ |
| $\gamma$ | 2.910 (0.209) | $2.604, 3.238$ |

Table 3: Parameter estimates from the model described in equation 7, reported to 3 decimal places. We report 90% confidence intervals for all of the parameter estimates by bootstrapping 100 iterations.

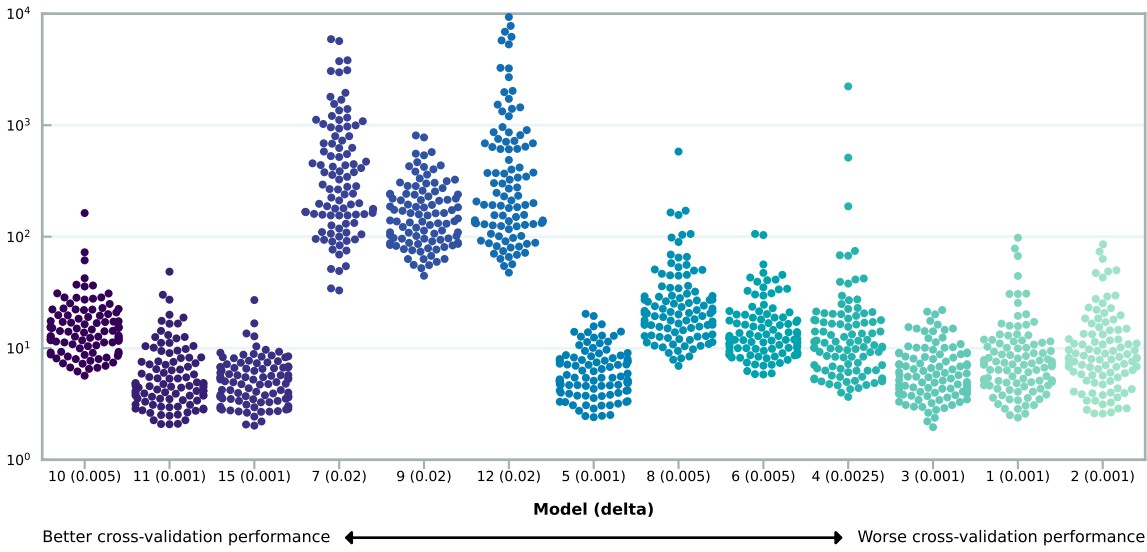

Figure 7: Contribution of the transformer in terms of Compute Equivalent Gain, as by introducing the structure in equation 7 to different model specifications.

# E  Dataset

## E.1  Performance measure and dataset

A key component of measuring progress in machine learning algorithms is the presence of a performance metric or benchmark. Since our focus is language modeling, token-level perplexity is a natural choice for this metric, for which we choose three benchmarks: WikiText-103, WikiText-2, and Penn Treebank. Note that WT2 and WT103 are both constructed from articles on Wikipedia. The two benchmarks share the same validation and testing set, while WikiText-103 has a much larger vocabulary and training set. In total, our dataset is constructed from 226 papers, from which we collect around 410 models that have reported token-level perplexity. Of these models, 370 contain sufficient information to be considered for analysis (perplexity, parameter size, publication date, and size of dataset).

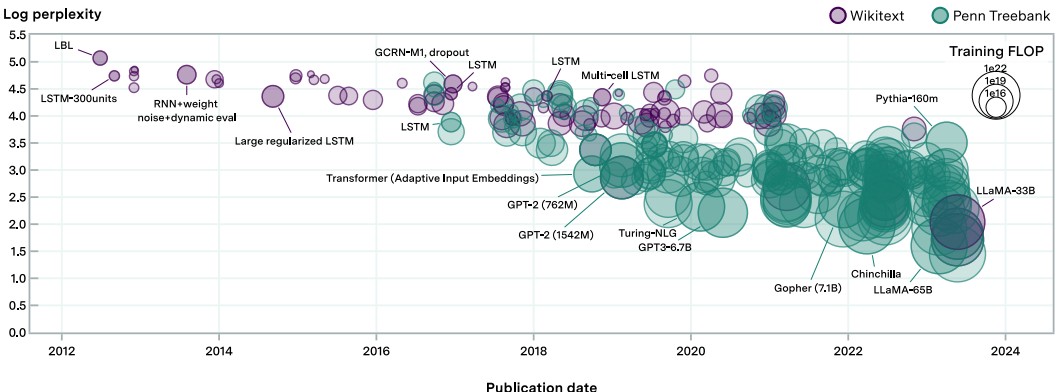

Figure 8: Log of perplexity of models used in our work, of over 231 language models analyzed in our work spanning over 8 orders of magnitude of compute, with each shape representing a model. The size of the shape is proportional to the compute used during training. Comparable perplexity evaluations are curated from the existing literature and from our own evaluations.

## E.2  Perplexity

A standard metric for LLM performance is the measured test perplexity on standard datasets. For a language model, this is defined in as the exponential of the cross-entropy $L$ between the model predictions and the test set, i.e. Perplexity $= e^L$.

We choose this metric for two primary reasons. First, this is a commonly reported measure of performance, which allows us to gather a large dataset for our analysis. Second, the simple relation between perplexity and cross entropy $L$ allows us to easily relate our model to neural scaling laws.

If a paper reports the perplexity of just one model, we collect that singular data point. However, when a paper presents multiple models, only those meeting any of the following criteria are included in our dataset:

1. The model is trained or evaluated on a different benchmark dataset. A model trained on two different datasets can be used as a reference to understand how perplexity metrics reported on different benchmarks relate to one another. This can be helpful e.g. for data imputation

2. The model is constructed with a drastically different parameter size, as such data inform the impact of scaling

3. The model has a significant difference in the algorithms used than other models in the paper

For papers presenting many (10 or more) models, we exclude some from our dataset to prevent possible bias from over-representing results from a few studies. We prioritize models with the lowest perplexity in their category, often highlighted in bold within tables. We also exclude minor algorithm alterations and ablations that do not impact the parameter count. In Appendix I, we take an alternative approach by including all models from each paper and explicitly modelling the autocorrelation

structure from results from the same paper. In doing so, we find results highly similar to those we present in the paper.

### E.2.1 Context length

Another consideration when analyzing algorithmic innovation in language models pertains to the context length. For one, measured perplexity on benchmarks can depend on the context length [Kaplan et al., 2020, Xiong et al., 2023]. Different systems may have been trained or evaluated using different context lengths, and this might make model perplexity scores less directly comparable.

One way to try and quantify the magnitude of this effect is to look at studies that compare the change in perplexity given a change in context length. For example, based on the scaling relations relating loss and context length from Xiong et al. [2023] and Kaplan et al. [2020], back-of-the-envelope calculations suggest that loss reductions each year due to increasing context length could be 10-60% as large as the loss reductions from algorithmic progress.

In particular, Xiong et al. [2023] finds a relation between context length $c$ and validation loss for different versions of the LLaMa 2 language model. We can estimate ballpark values for context length over time based on data from de Vries [2023], and use this to roughly estimate how expanding context lengths has decreased loss over time (e.g. a decrease of a few percent per year). We can then compare the magnitude of this effect to the contribution of algorithmic progress to decreases in loss, and we typically arrive at values between 10-60% of the overall algorithmic progress contribution.[14] We perform a similar analysis with the scaling relation described in Kaplan et al. [2020], with similar results. If this rough calculation is correct, it suggests that increasing context length may have been a fairly important dimension along which algorithms have been improving.

On top of measured reductions to averaged perplexity per token, one might also consider the increasing ability of language models to perform long-context tasks to be largely downstream of algorithmic progress in itself. Being able to handle long contexts is a key motivator for several recent algorithmic innovations [Liu et al., 2024, Gu and Dao, 2023, Gemini Team, 2024], and this has likely grown very substantially since the introduction of FlashAttention [Dao et al., 2022, de Vries, 2023]. We consider this to be an important avenue for further investigation.

### E.2.2 Tokenization

One way to get a sense of the impact of tokenization on measured doubling times is to introduce a fixed effect that depends on the benchmark vocabulary size into our preferred model. In particular, we introduce an irreducible loss term to Equation 8, of the form $\gamma \log(\text{vocabulary size})$.

As with the rest of our analysis, we perform bootstraps to obtain a distribution over estimated doubling times, and further fold this model into our cross validation analysis. In particular, this model predicts a median effective compute doubling time of 5.5 months, with a 90% confidence interval of 2.6 to 11.8 months. In cross validation, this model performs essentially as well as our preferred model, with a MSE loss of 0.046–0.048 in both cases (depending on the regularization strength). Both of these results are very much in line with the results from our main model, lending some weight to the view that differences in tokenization schemes used in practice do not substantially change our core results.

One possible reason for this is that in practice, the tokenization schemes used for evaluating language models on the considered benchmarks (i.e. WT103, WT2 and PTB) are typically highly similar, and so including a contribution from vocabulary size has only a limited effect within each dataset. For example, typical tokenizers for PTB, WT2 and WT103 have vocabularies of roughly 10k, 22k, and 268k tokens respectively. We illustrate this in Figure 9, where for each benchmark, the majority of vocabularies fall into the same histogram bin.

### E.2.3 Inconsistencies in perplexity evaluations

Inconsistencies in benchmark evaluations is a well-known issue in the machine learning community. These issues can introduce noise (if inconsistencies are 'random'), or bias if they systematically change over time. We curated data so that models were evaluated in roughly comparable ways, but often the precise details of evaluations were lacking, so that we could not verify the precise

---

[14]This of course depends on variables like how quickly context lengths have expanded over time—details of this calculation can be found in this spreadsheet.

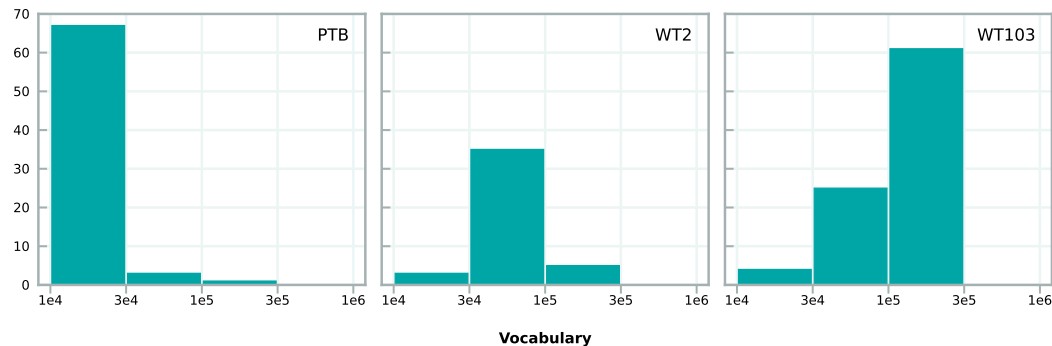

Figure 9: Histogram showing the most common vocabulary sizes for models in our dataset, separated by benchmark.

evaluation procedure. However, there are many other subtleties with evaluation setups that may also cause perplexity results to differ, such as pretraining data, test-time adaptation [Takase et al., 2018], tokenization schemes, strides, and text preprocessing.

| Variation source | Significance |
|---|---|
| Training set | Early models in our dataset are often exclusively trained on the benchmark training set before evaluation, whereas later models are generally pretrained on a larger pretraining dataset. Since the majority of our dataset involves the latter type of model, we expect this effect to be minor. The direction of this effect is somewhat ambiguous: the training set data is likely 'closer' to the test distribution relative to other internet text-corpuses, so not training on the relevant training distribution could yield lower performance at fixed budgets. |
| | A further subtlety is that large text corpuses often, but not always, contain Wikipedia data. Pretraining on such distributions could yield larger gains than otherwise, which could be attributed to algorithmic progress. A small number of existing results illustrate this effect. For example, in a 1.3B GPT-3 reimplementation, including Wikipedia and other high-quality data in fixed-size pretraining reduced WikiText perplexity from 8.27 to 5.59 [Gao et al., 2020, Table 3]. In our dataset, this effect is likely to show up as a small number of models receiving worse-than-otherwise perplexities in WikiText, around the time of the transition to large-scale pretraining but before inclusion of Wikipedia data became common. This is true of GPT-2, for example. |
| Word vs sub-word tokenization | Changing to a sub-word vocabulary can reduce perplexity substantially in an otherwise unchanged architecture, for example by $\sim$30% in a pre-GPT-2 LSTM [Wang et al., 2019]. In our dataset this is likely to show up as a one-time change, potentially exaggerating the algorithmic improvement in language models around GPT onwards. |
| Preprocessing and runtime adaptation | Different preprocessing of data can affect results significantly. For example, inverting word-level tokenization artifacts in WikiText-103 improved perplexity by $\sim$10% [Radford et al., 2019]. Runtime adaptation sometimes has similarly large effects [Alon et al., 2022]. We expect that these sources of variation mostly act as noise in our dataset, although if these improved over time, they might inflate estimates of algorithmic efficiency. |
| Stride length | The move to larger models led to evaluations using larger stride for lower computational cost. This can increase perplexity, but only on the order of $\sim$10% at realistic settings [Hugging Face, 2023]. We believe this should act as another source of variation, but without a strong influence on our overall findings. |

Table 4: Sources of variation and their significance in language modelling evaluations.

Overall, we do not see these issues substantially undermining our results. Our mainline estimates imply that 1 year of pre-training algorithmic progress amounts to a reduction of perplexity of around

|  | Estimate | 90% CI |
|---|---|---|
| $\alpha_{\text{const}}$ | 1.126 (0.060) | $1.008, 1.217$ |
| $\alpha_{\text{const,PTB}}$ | 0.172 (0.058) | $0.070, 0.219$ |
| $\alpha_{\text{const,WT2}}$ | 0.086 (0.061) | $-0.003, 0.187$ |
| $\alpha_{\text{year}}$ | 0.009 (0.008) | $-0.009, 0.016$ |
| $\alpha_{\text{param}}$ | 0.052 (0.008) | $0.040, 0.067$ |
| $\beta_{\text{const}}$ | 0.349 (0.125) | $0.221, 0.595$ |
| $\beta_{\text{const,PTB}}$ | 0.002 (0.086) | $-0.001, 0.120$ |
| $\beta_{\text{const,WT2}}$ | 0.171 (0.131) | $-0.000, 0.412$ |
| $\beta_{\text{year}}$ | 0.043 (0.022) | $0.020, 0.091$ |
| $\beta_{\text{data}}$ | 0.037 (0.017) | $0.015, 0.060$ |

Table 5: Parameter estimates for our main model, fitted to a dataset of models trained on the training set of WikiText or PTB, but not the corresponding test sets. We report $90\%$ confidence intervals for all of the parameter estimates by bootstrapping 100 iterations.

10%, which is a much larger reduction than seems plausible to explain by changes in the average year-to-year evaluation procedures.

Given that test perplexity on these datasets has persisted as a standard measure of language modeling performance in the literature, we expect that differences in perplexity will broadly reflect genuine underlying differences in model capabilities.

We further consider one particular source of variation from Table 4 in more detail, namely the training set. To do this, we went through all the models in our dataset and classified them into one of three categories: (1) not trained on either the training or test sets of WikiText, (2) trained on the training set but not the test set of WikiText, or (3) trained on both the training and test sets of WikiText.[15] We then repeated this procedure for PTB.

Based on this classification, around 2% of models fall into the first category, around 85% of models are in the second category, and the remaining models are in the third category. To test whether our results are robust to this change in categories, we fit our model to models only in the second category. This yields effective compute doubling times of 5.7 months in the median case, with a 90% confidence interval of [1.7, 12.6] months, consistent with our core results. The parameter estimates and associated uncertainties are listed in Table 5.

### E.3 Dataset Size & Epochs

In reporting training dataset size, we generally record the size of pre-training datasets as well fine-tuning datasets if the model in question has been fine-tuned. The number of epochs is usually sourced from the papers which describe the language model in question, but if not provided, we estimate it via

$$\texttt{num\_epochs} = \frac{\texttt{context\_length\_tokens} \cdot \texttt{batch\_size} \cdot \texttt{training\_steps}}{\texttt{pretrain\_tokens}}.$$

We adjust our epoch and dataset size calculations accordingly for models like GPT-Neo and Pythia, which report the effective number of tokens seen in training.

### E.4 Parameter Size

We use the reported parameter size value if it is stated in the paper. We impute the parameter size from the previous models for papers that do not specify parameter size but follow known, state-of-the-art

---

[15]Note that the test sets for WT103 and WT2 are the same, so we group them together here to consider "WikiText".

models. Otherwise, we rely on other papers referring to the model's parameter size or manually compute parameters for certain RNN and LSTM models based on provided details about the model architecture (e.g. the number and size of hidden layers).

### E.5 Inclusion and exclusion criteria

We exclude several models in the dataset from the analysis based on whether they meet any of the following primary criteria: (1) use of a retrieval mechanism, (2) use of model compression or pruning, (3) use of neural architecture search, (4) use of teacher-learner or knowledge distillation mechanisms, (5) use of cache models. These models are excluded because we expect these models to exhibit significantly different scaling behaviors from other models we analyze. In particular, they can substantially change the ratio of parameters and data that would "optimally" minimize loss given some compute budget.

### E.6 Dataset review

In a previous version of this paper, our core results had a narrower confidence interval as well as slightly longer median estimates.[16] One reason for this change is that reviewed the data a second time and fixed any identified errors – while this changed our results, it was well within the uncertainty ranges we had previously identified in our original paper.

That said, we further decided to modify our model aggregation approach to better represent cross-model uncertainty. In particular, rather than aggregating over the top 10 model-delta pairs in cross validation, we instead aggregate across different model numbers (with the regularization strength $\delta$ that yields the best cross validation performance).

## F Quantifying training data $D$

One important degree of freedom in the modeling process is how to define the "training data". In particular, there we consider three possible definitions:

1. **Training dataset size:** The number of tokens in the dataset on which the language model was trained. This definition ignores the possibility of improvements in performance from training for multiple epochs, and is the approach taken in Erdil and Besiroglu [2022].

2. **Tokens seen:** The total number of tokens seen during the course of training a language model—this is the definition adopted in Hoffmann et al. [2022]. This is equivalent to the training dataset size if the model is trained for one epoch on the training set, which is fairly common practice but not totally ubiquitous. In fact, it is possible that more recent models are being pushed towards training with multiple epochs on the same data—e.g. GPT-4 was reportedly trained for 2 epochs on text and 4 epochs on code [Patel and Wong, 2023]. In cases where the tokens seen is not directly reported in paper, we estimate it via tokens seen $\approx$ num. epochs $\times$ training dataset size.

3. **Tokens seen with diminishing returns:** One problem with the previous approach is that seeing data repeatedly may yield diminishing returns. Muennighoff et al. [2023] find that the benefits in loss reduction drop significantly when training on more than 4 epochs.

In order to test the robustness of our most important result (compute doubling times) to this degree of freedom, we repeat our doubling times analysis as in section 3.1 but account for the number of training epochs. We then replace $D$ either with tokens seen (epochs times training dataset size) or tokens seen with diminishing returns. For each case we consider two possibilities—either we drop datapoints for which the epoch number is unknown (this drops around 100 datapoints), or we impute the epoch number as 1. We report our core parameter estimates in Tables 7 and 8, and our doubling time estimates in Table 6.

From the results in Table 6, we see that there is substantial overlap between model estimates, but the estimates are highly noisy. The most similar are the original model based on dataset size, and model

---

[16]In particular, our core model had a median doubling time estimate of 8.4 months [4.5, 14.3], and our aggregate model had a median of 7.8 months [1.5, 17.6]. In contrast, our core model now has a central estimate of 6.1 months [2.4, 14.8] and our aggregate model has a median estimate of 7.5 months [0.9, 24.6].

| Definition | $C_{\text{eff}}$ doubling times (months) |
|---|---|
| Dataset size (section 3.1) | [2.6, 6.1, 14.8] |
| Tokens seen | [0.9, 3.0, 22.0] |
| Tokens seen + impute | [-7.3, 1.7, 20.2] |
| Tokens seen w. dim. returns | [0.9, 3.4, 10.9] |
| Tokens seen w. dim. returns + impute | [2.4, 5.8, 16.4] |

Table 6: Estimated effective compute doubling times using the core model (Equation 8), using three different definitions of "training data". Numbers in the square brackets correspond to the [2.5th, 50th, and 97.5th percentile] after bootstrapping 100 times.

with tokens seen with diminishing returns and imputed epochs, where the latter places slightly more weight on longer doubling times.

In the case of tokens seen with imputed epochs, the model predicts no statistically significant improvement in compute efficiency. On the other hand we do not observe this without imputation, which potentially suggests that our imputation strategy (i.e. assuming 1 epoch of training when epoch counts are unknown) is suboptimal under this definition of training data.

That said, there are strong reasons to be sceptical of using "tokens seen" as a definition given that diminishing returns to additional training epochs are indeed observed in practice. Indeed, some papers report training for between 100-1000 epochs, which would result in fairly large variations in dataset size estimates if diminishing returns are not considered. We ultimately opt to quantify training data in terms of dataset size for simplicity, although a definition based on diminishing returns is also feasible.

| | Estimate | 90% CI |
|---|---|---|
| $\alpha_{\text{const}}$ | 0.215 (0.416) | 0.088, 1.342 |
| $\alpha_{\text{const,PTB}}$ | −0.000 (0.210) | −0.471, 0.205 |
| $\alpha_{\text{const,WT2}}$ | 0.014 (0.183) | −0.043, 0.462 |
| $\alpha_{\text{year}}$ | −0.090 (0.040) | −0.101, 0.011 |
| $\alpha_{\text{param}}$ | 0.168 (0.036) | 0.077, 0.193 |
| $\beta_{\text{const}}$ | 1.299 (0.406) | 0.001, 1.396 |
| $\beta_{\text{const,PTB}}$ | 0.152 (0.210) | 0.001, 0.732 |
| $\beta_{\text{const,WT2}}$ | 0.116 (0.250) | −0.468, 0.200 |
| $\beta_{\text{year}}$ | 0.070 (0.030) | 0.020, 0.113 |
| $\beta_{\text{data}}$ | 0.018 (0.014) | 0.012, 0.055 |

Table 7: $D =$ tokens seen, without imputation of epochs.

| | Estimate | 90% CI |
|---|---|---|
| $\alpha_{\text{const}}$ | 0.914 (0.391) | −0.000, 1.175 |
| $\alpha_{\text{const,PTB}}$ | −0.000 (0.188) | −0.437, 0.133 |
| $\alpha_{\text{const,WT2}}$ | 0.047 (0.166) | −0.076, 0.359 |
| $\alpha_{\text{year}}$ | −0.020 (0.035) | −0.101, 0.011 |
| $\alpha_{\text{param}}$ | 0.090 (0.035) | 0.070, 0.172 |
| $\beta_{\text{const}}$ | 0.779 (0.285) | 0.463, 1.316 |
| $\beta_{\text{const,PTB}}$ | 0.233 (0.176) | 0.000, 0.467 |
| $\beta_{\text{const,WT2}}$ | 0.122 (0.194) | −0.189, 0.313 |
| $\beta_{\text{year}}$ | 0.071 (0.026) | 0.028, 0.114 |
| $\beta_{\text{data}}$ | 0.026 (0.012) | 0.009, 0.047 |

Table 8: $D =$ tokens seen with diminishing returns and without imputation of epochs.

Parameter estimates from the model described in equation 8, to 3 decimal places. We report 90% confidence intervals for all of the parameter estimates by bootstrapping 100 iterations.

# G   Doubling times via optimal scaling

In the main paper we calculated doubling times for effective compute based on a closed form solution for the doubling times, given by Equation 6. However, this equation was derived simply by considering changes in $D_{\text{eff}}$, and it is not clear *a priori* whether or not the effective compute is optimally allocated between $N_{\text{eff}}$ and $D_{\text{eff}}$ to minimize the cross entropy loss. In addition, calculating compute efficiency doubling times is less straightforward for models which also include changing scale exponents $\alpha_{\text{param}}$ and $\beta_{\text{data}}$ (i.e. models 14 and 15 in our cross validation analysis, see J). We

thus supplement our previous calculation with an alternative approach, which instead enforces the condition of compute-optimal scaling.

We approach this in two stages:

1. First, we calculate the reduction in cross entropy loss $\Delta L$ given a doubling in compute budgets and under compute-optimality. Let the initial compute budget be $C$. We solve an optimization problem

$$L_1 = \min_{(N,D)} L(N,D), \tag{9}$$

   subject to the constraint $C = 6ND$ described in Hoffmann et al. [2022], which is solved with values $N_1^*$ and $D_1^*$. We then perform the same optimization problem but with a budget constraint of $2C = 6ND$, yielding a corresponding cross entropy loss of $L_2$. The reduction in cross entropy loss is given by $\Delta L = L_2 - L_1 \leq 0$.

2. We then estimate the years of algorithmic progress that would be needed to achieve this same reduction $\Delta L$. In particular, the optimization problem is

$$\min_{\delta \in \mathbb{R}^+} f(\delta), \tag{10}$$

   where we have

$$
\begin{aligned}
f = \big| \exp[\alpha'_{\text{const}} - \alpha_{\text{year}}(Y + \delta - Y_0) - \alpha_{\text{param}} \log(N/N_0)] \\
+ \exp[\beta'_{\text{const}} - \beta_{\text{year}}(Y + \delta - Y_0) - \beta_{\text{data}} \log(D/D_0)] - L_2 \big|
\end{aligned} \tag{11}
$$

   . Here $\delta$ can be interpreted as a doubling time for effective compute due to algorithmic progress in pre-training. $\alpha'_{\text{const}} = \alpha_{\text{const}} + \alpha_{\text{const,PTB}} x_{\text{PTB}} + \alpha_{\text{const,WT2}} x_{\text{WT2}}$ and $\beta'_{\text{const}} = \beta_{\text{const}} + \beta_{\text{const,PTB}} x_{\text{PTB}} + \beta_{\text{const,WT2}} x_{\text{WT2}}$, where $x_{\text{PTB}}$ and $x_{\text{WT2}}$ are dummy variables for PTB and WT2 respectively.

We apply this approach over 100 bootstraps of our dataset, yielding a median doubling time of 6.1 months, and a 90% confidence interval of 3.3 to 11.3 months. This is very similar to the doubling times estimated using the closed-form approach discussed in section 3.1, which also has a median of 6.1 months [3.3, 11.3] for model 7.

## H   Irreducible loss

Our main model for estimating doubling times does not estimate the irreducible loss. This is in part due to empirical difficulties encountered in estimating plausible values for this term, and in part because the inclusion of this term does not have a bearing on our estimates of the rate of algorithmic improvements. Since the latter is the focus of our paper, we decided to move forward with the outlined model without irreducible loss estimates. However, we caution against overinterpretation of these results—for instance, our parameter estimates are not reliable enough to strongly inform how to scale models compute-optimally (although they can be somewhat illustrative).

The focus of this section is to justify the robustness of our core doubling time results to this omitted variable in our model. In particular, we fit a model which incorporates estimates of the irreducible loss and show that the doubling times remain in line with our previous findings. The model we consider is

$$\hat{L} = \gamma' + \exp(\alpha'_{\text{const}} - \alpha_{\text{year}}(Y - Y_0) - \alpha_{\text{param}} \log N/N_0) + \exp(\beta'_{\text{const}} - \beta_{\text{year}}(Y - Y_0) - \beta_{\text{data}} \log D/D_0), \tag{12}$$

where $\gamma' = \gamma + \gamma_{PTB} x_{PTB} + \gamma_{WT2} x_{WT2}$, and the rest of the model is defined in the same way as in Section 3.1. When we fit this model, we obtain a compute efficiency doubling time of 5.4 months, with a 90% confidence interval of 2.4 to 10.8 months, consistent with the estimates from our primary model. Our core parameter estimates are shown in Table 9.

## I   Autocorrelation

Our dataset was constructed by searching for papers with models that reported perplexity data on WT103, PTB, or WT2. In our initial data collection, we sometimes also included multiple models

|  | Estimate | 90% CI |
|---|---|---|
| $\gamma$ | 0.000 (0.045) | $-0.007, 0.070$ |
| $\gamma_{\text{PTB}}$ | 0.214 (0.178) | $0.000, 0.460$ |
| $\gamma_{\text{WT2}}$ | $-0.111$ (0.147) | $-0.409, 0.001$ |
| $\alpha_{\text{const}}$ | 0.833 (0.246) | $0.627, 1.154$ |
| $\alpha_{\text{const,PTB}}$ | 0.001 (0.097) | $-0.116, 0.134$ |
| $\alpha_{\text{const,WT2}}$ | 0.102 (0.103) | $-0.118, 0.252$ |
| $\alpha_{\text{year}}$ | $-0.005$ (0.022) | $-0.034, 0.013$ |
| $\alpha_{\text{param}}$ | 0.085 (0.019) | $0.057, 0.099$ |
| $\beta_{\text{const}}$ | 0.882 (0.227) | $0.438, 1.091$ |
| $\beta_{\text{const,PTB}}$ | 0.076 (0.161) | $-0.025, 0.324$ |
| $\beta_{\text{const,WT2}}$ | 0.113 (0.170) | $-0.000, 0.476$ |
| $\beta_{\text{year}}$ | 0.044 (0.018) | $0.026, 0.079$ |
| $\beta_{\text{data}}$ | 0.028 (0.009) | $0.019, 0.050$ |

Table 9: Parameter estimates from the model described in equation 12, reported to 3 decimal places. We report 90% confidence intervals for all of the parameter estimates by bootstrapping 100 iterations.

originating from the same paper. In some extreme instances, more than ten models from a single paper were included; for example, we incorporated 14 models from the paper "OPT: Open Pre-trained Transformer Language Models." This poses concerns of autocorrelation, which might for instance result in us underestimating the uncertainty in our individual parameter estimates.

In the main body of the paper we approached this issue by retaining only three models per paper in our analysis, which resulted in the exclusion of approximately 35 models. Here we consider an alternative approach for addressing autocorrelation, where we explicitly quantify the correlations between models from the same paper. We then use this information to establish a multivariate normal likelihood function, which we maximize to obtain parameter estimates.

First, let us define the residuals as $\epsilon = \mathbf{x} - \hat{\mathbf{x}}$. The original loss function that we are using is the mean squared error, i.e. where the loss is given by $E[\epsilon^T \epsilon]$. We want to modify our approach so that we take into account the correlations between different datapoints. The approach we take is to take an approach similar to generalized least squares and multiplicative attention—rather than consider just $\epsilon^T \epsilon$, we consider $\epsilon^T P \epsilon$, where $P$ is a correlation matrix.

We do this using a maximum-likelihood approach, where $\epsilon^T \Sigma \epsilon$ placed in a multivariate normal distribution, given by

$$f(\epsilon; \theta) = \frac{1}{\sqrt{(2\pi)^k \det(\Sigma)}} \exp\left(-\frac{1}{2}\epsilon^T \Sigma^{-1} \epsilon\right), \tag{13}$$

where $\Sigma$ is a covariance matrix for the data. Our goal is to choose the appropriate parameters $\theta$ such that the resulting $\epsilon = \mathbf{x} - \bar{\mathbf{x}}$ maximizes this distribution. This includes the original parameters in the model from equation 3 as well as the correlation $\rho$ between models from the same paper. We define the loss function as the negative of the logarithm of this distribution (dropping constants which do not matter for the resulting minimization problem):

$$l(\theta) = \frac{1}{2} \log \det \Sigma + \frac{1}{2}\epsilon^T \Sigma^{-1} \epsilon. \tag{14}$$

In order to apply this to our data we need to specify the structure of the covariance matrix $\Sigma = \sigma^{2n} P$ (and thus $\Sigma^{-1} = \sigma^{-2n} P^{-1}$. We can accordingly write the loss function as

$$l(\theta) = \frac{1}{2} \log \det P + \frac{n}{2} \log \sigma^2 + \frac{1}{2\sigma^{2n}}\epsilon^T P^{-1} \epsilon. \tag{15}$$

We are assuming that the models from different papers are uncorrelated, and that different models from the same paper have a correlation coefficient of $\rho$. If we order the models such that all the models from the same paper form a contiguous range of indices, then the correlation matrix looks block diagonal, where each block has 1s on the diagonal and $\rho$ for the off-diagonal terms. The matrix elements that are not in blocks are all zero. For example, one example correlation matrix is:

$$
\begin{pmatrix}
1 & \rho & 0 & 0 & 0 & 0 \\
\rho & 1 & 0 & 0 & 0 & 0 \\
0 & 0 & 1 & \rho & \rho & \rho \\
0 & 0 & \rho & 1 & \rho & \rho \\
0 & 0 & \rho & \rho & 1 & \rho \\
0 & 0 & \rho & \rho & \rho & 1
\end{pmatrix}
\tag{16}
$$

As we can see, we need to determine both the inverse and the determinant of the correlation matrix $P$ in order to calculate the negative log-likelihood. While this can be done using standard libraries, the matrices that we are considering here are quite sparse, and thus it is more efficient to simplify our calculations here. We detail the calculations in sections I.2 and I.3.

## I.1   Results

To determine confidence intervals, we bootstrap this model based on clustering over 100 iterations, in similar fashion to the main results. This yields a median doubling time of 4.5 months, with a 90% confidence interval of 1.5 to 9.9 months, which is consistent with our core estimates. In practice $\rho$ is typically on the order of 0.45, which suggests that the degree of autocorrelation is not very strong. We report our parameter estimates in Table 10.

|  | Estimate | 90% CI |
|---|---|---|
| $\alpha_{\text{const}}$ | $0.355$ $(0.753)$ | $-0.802, 1.513$ |
| $\alpha_{\text{const,PTB}}$ | $0.058$ $(0.929)$ | $-1.009, 0.267$ |
| $\alpha_{\text{const,WT2}}$ | $0.125$ $(0.203)$ | $-0.133, 0.582$ |
| $\alpha_{\text{year}}$ | $-0.052$ $(0.068)$ | $-0.163, 0.028$ |
| $\alpha_{\text{param}}$ | $0.120$ $(0.051)$ | $0.068, 0.214$ |
| $\beta_{\text{const}}$ | $1.189$ $(0.868)$ | $-0.707, 1.471$ |
| $\beta_{\text{const,PTB}}$ | $0.111$ $(1.242)$ | $-0.109, 1.623$ |
| $\beta_{\text{const,WT2}}$ | $0.091$ $(1.247)$ | $-2.397, 0.909$ |
| $\beta_{\text{year}}$ | $0.060$ $(0.029)$ | $0.016, 0.087$ |
| $\beta_{\text{data}}$ | $0.020$ $(0.041)$ | $0.011, 0.034$ |

Table 10: Parameter estimates from the model described in equation 8, but estimated using the clustering approach described in equation 15. Estimates are rounded to 3 decimal places. We report 90% confidence intervals for all of the parameter estimates by bootstrapping 100 iterations.

## I.2   Determinant

In order to obtain the overall determinant for $P$ we first work this out for a single block. In particular, the determinant of each $(n + 1) \times (n + 1)$ block $B_n$ is given by

$$
\det B_{n+1} = (1 - \rho)^n (1 + n\rho).
\tag{17}
$$

We prove this by considering the associated matrix directly:

$$
B_{n+1} = \begin{pmatrix}
1 & \rho & \cdots & \rho & \rho \\
\rho & 1 & \cdots & \rho & \rho \\
\vdots & & \ddots & & \vdots \\
\rho & \rho & \cdots & 1 & \rho \\
\rho & \rho & \cdots & \rho & 1
\end{pmatrix}
\tag{18}
$$

Since $\det B_{n+1}$ is unchanged under the elementary row operation of adding a multiple of one row to another, we write

$$
\det B_{n+1} = \det_{(n+1)\times(n+1)} \begin{pmatrix}
1 & \rho & \cdots & \rho & \rho \\
0 & 1-\rho^2 & \cdots & \rho-\rho^2 & \rho-\rho^2 \\
\vdots & & \ddots & & \vdots \\
0 & \rho-\rho^2 & \cdots & 1-\rho^2 & \rho-\rho^2 \\
0 & \rho-\rho^2 & \cdots & \rho-\rho^2 & 1-\rho^2
\end{pmatrix}.
\tag{19}
$$

We can thus simplify the determinant as

$$
\det B_{n+1} = (1-p)^n \det_{n\times n} \begin{pmatrix}
1+\rho & \rho & \cdots & \rho & \rho \\
\rho & 1+\rho & \cdots & \rho & \rho \\
\vdots & & \ddots & & \vdots \\
\rho & \rho & \cdots & 1+\rho & \rho \\
\rho & \rho & \cdots & \rho & 1+\rho
\end{pmatrix}.
\tag{20}
$$

To evaluate the determinant of this matrix we follow a similar procedure, where we eliminate most of the elements in the first column. We then repeat this process as the resulting matrix becomes smaller and smaller, until we reach a trivial case. The $k$th step in this iterative procedure has the following structure (where $k = 0, 1, \ldots, n-1, n$):

$$
\det B_{n+1} = (1-p)^n(1+k\rho) \det_{(n-k)\times(n-k)} \begin{pmatrix}
\frac{1+(k+1)\rho}{1+k\rho} & \frac{\rho}{1+k\rho} & \cdots & \frac{\rho}{1+k\rho} & \frac{\rho}{1+k\rho} \\
\frac{\rho}{1+k\rho} & \frac{1+(k+1)\rho}{1+k\rho} & \cdots & \frac{\rho}{1+k\rho} & \frac{\rho}{1+k\rho} \\
\vdots & & \ddots & & \vdots \\
\frac{\rho}{1+k\rho} & \frac{\rho}{1+k\rho} & \cdots & \frac{1+(k+1)\rho}{1+k\rho} & \frac{\rho}{1+k\rho} \\
\frac{\rho}{1+k\rho} & \frac{\rho}{1+k\rho} & \cdots & \frac{\rho}{1+k\rho} & \frac{1+(k+1)\rho}{1+k\rho}
\end{pmatrix}
\tag{21}
$$

We now substract $\frac{\rho}{1+(k+1)\rho}$ times the first row from all other rows. In the first column, all except the first row thus becomes zeros, and there are two main cases we need to consider for the other elements. In the first case, for diagonal elements, we have we have

$$
\frac{1+(k+1)\rho}{1+k\rho} - \frac{\rho}{1+k\rho}\frac{\rho}{1+(k+1)\rho} = \frac{[1+(k+1)\rho]^2 - \rho^2}{[1+k\rho][1+(k+1)\rho]}
\tag{22}
$$

$$
= \frac{1+(k+2)\rho}{1+(k+1)\rho}.
\tag{23}
$$

In the second case, for off-diagonal elements, we instead have

$$
\frac{\rho}{1+k\rho} - \frac{\rho}{1+k\rho}\frac{\rho}{1+(k+1)\rho} = \frac{\rho}{1+k\rho}\left[\frac{1+(k+1)\rho-\rho}{1+(k+1)\rho}\right]
\tag{24}
$$

$$
= \frac{\rho}{1+(k+1)\rho}.
\tag{25}
$$

Thus we have that

$$
\det B_{n+1} = (1-p)^n(1+k\rho) \det_{(n-k)\times(n-k)} \begin{pmatrix}
\frac{1+(k+1)\rho}{1+k\rho} & \frac{\rho}{1+k\rho} & \cdots & \frac{\rho}{1+k\rho} & \frac{\rho}{1+k\rho} \\
0 & \frac{1+(k+2)\rho}{1+(k+1)\rho} & \cdots & \frac{\rho}{1+(k+1)\rho} & \frac{\rho}{1+(k+1)\rho} \\
\vdots & & \ddots & & \vdots \\
0 & \frac{\rho}{1+(k+1)\rho} & \frac{1+(k+2)\rho}{1+(k+1)\rho} & \frac{\rho}{1+(k+1)\rho} & \\
0 & \frac{\rho}{1+(k+1)\rho} & \cdots & \frac{\rho}{1+(k+1)\rho} & \frac{1+(k+2)\rho}{1+(k+1)\rho}
\end{pmatrix}
\tag{26}
$$

$$= (1-p)^n(1+(k+1)\rho) \det_{(n-(k+1))\times(n-(k+1))} \begin{pmatrix} \frac{1+(k+2)\rho}{1+(k+1)\rho} & \frac{\rho}{1+(k+1)\rho} & \cdots & \frac{\rho}{1+(k+1)\rho} & \frac{\rho}{1+(k+1)\rho} \\ \frac{\rho}{1+(k+1)\rho} & \frac{1+(k+2)\rho}{1+(k+1)\rho} & \cdots & \frac{\rho}{1+(k+1)\rho} & \frac{\rho}{1+(k+1)\rho} \\ \vdots & & \ddots & & \vdots \\ \frac{\rho}{1+(k+1)\rho} & \frac{\rho}{1+(k+1)\rho} & \cdots & \frac{1+(k+2)\rho}{1+(k+1)\rho} & \frac{\rho}{1+(k+1)\rho} \\ \frac{\rho}{1+(k+1)\rho} & \frac{\rho}{1+(k+1)\rho} & \cdots & \frac{\rho}{1+(k+1)\rho} & \frac{1+(k+2)\rho}{1+(k+1)\rho} \end{pmatrix}.$$
(27)

If we repeat this argument $n$ times (going from an $(n+1) \times (n+1)$ matrix to the trivial case of a $n \times n$ matrix, then we conclude that

$$\det B_{n+1} = (1-\rho)^n(1+n\rho),$$
(28)

as desired. Now to work out the overall determinant of $P$, we simply have to multiply the determinants of the individual blocks, and we are done.

## I.3 Inverse

We now want to determine the third term in $l(\theta)$, i.e. $\frac{1}{2\sigma^{2n}}\epsilon^T P^{-1}\epsilon$. One way to reduce the computational cost of this calculation is to take advantage of $P$ being block diagonal, where the inverse $P^{-1}$ can be determined by inverting the individual blocks. That is, for blocks $B_i$ the inverse of the correlation matrix is

$$P^{-1} = \begin{pmatrix} B_1^{-1} & 0 & \cdots & 0 & 0 \\ 0 & B_2^{-1} & \cdots & 0 & 0 \\ \vdots & & \ddots & & \vdots \\ 0 & 0 & \cdots & B_{n-1}^{-1} & 0 \\ 0 & 0 & \cdots & 0 & B_n^{-1} \end{pmatrix}.$$
(29)

To invert a single block $B$, first observe that each block $B$ is such that the elements are

$$B_{ij} = \begin{cases} 1 & \text{if } i = j, \\ \rho & \text{otherwise.} \end{cases}$$
(30)

Now let $D$ be the diagonal matrix with diagonal entries $D_{ii} = 1 - \rho$. Then we have that $B = D + \rho vv^T$, where $v$ is a vector of ones. We can now calculate the inverse $B^{-1}$ by application of the Sherman-Morrison formula. In particular we have

$$B^{-1} = (D + \rho vv^T)^{-1} = D^{-1} - \frac{\rho D^{-1}vv^T D^{-1}}{1 + \rho v^T D^{-1}v}.$$
(31)

Since the inverse of $D$ is just $D_{ii}^{-1} = \frac{1}{1-\rho}$, assuming that the correlation $\rho \neq 1$. To simplify notation, we define $c = \frac{1}{\rho} + \frac{n}{1-\rho}$. We then have

$$B_{ij}^{-1} = \begin{cases} \frac{1}{1-\rho} - \frac{1}{c(1-\rho)^2} & i = j \\ -\frac{1}{c(1-\rho)^2} & i \neq j \end{cases},$$
(32)

and calculating the associated quadratic form $\frac{1}{2\sigma^{2n}}\epsilon^T P^{-1}\epsilon$ follows trivially.

## J Cross validation for model choice

We determined our primary model for estimating doubling times using formal model selection procedures. In particular, we considered a range of 15 candidate model numbers and a range of possible regularization strengths $\delta \in \{0, 0.001, 0.0025, 0.005, 0.01, 0.02\}$. We consider each "model" as a *pair*, consisting of a particular model number and a regularization strength $\delta$. We then perform leave-one-out cross validation on all of these models. Here we show the results of this analysis.

Our candidate models are defined by varying three degrees of freedom:

1. Which parameters to make benchmark-specific (e.g. having three separate parameters for the exponent on training dataset size, one for each of WT103, PTB and WT2).

2. Whether to explicitly model algorithmic progress in parameters $N$, data $D$, or both.

In general we do not include the irreducible loss in the model (i.e. we usually set $E = 0$, in the equation 3)—this is described in more detail in section H. In Tables 11 and 12 we list the definitions of all models that we looped through in leave-one-out cross validation. For this, we first define some common terms for simplification:

$$\alpha'_{\text{const}} = \alpha_{\text{const}} + \alpha_{\text{const,PTB}} x_{\text{PTB}} + \alpha_{\text{const,WT2}} x_{\text{WT2}}$$

$$\alpha'_{\text{year}} = \alpha_{\text{year}} + \alpha_{\text{year,PTB}} x_{\text{PTB}} + \alpha_{\text{year,WT2}} x_{\text{WT2}}$$

$$\alpha'_{\text{param}} = \alpha_{\text{param}} + \alpha_{\text{param,PTB}} x_{\text{PTB}} + \alpha_{\text{param,WT2}} x_{\text{WT2}}$$

$$\beta'_{\text{const}} = \beta_{\text{const}} + \beta_{\text{const,PTB}} x_{\text{PTB}} + \beta_{\text{const,WT2}} x_{\text{WT2}}$$

$$\beta'_{\text{year}} = \beta_{\text{year}} + \beta_{\text{year,PTB}} x_{\text{PTB}} + \beta_{\text{year,WT2}} x_{\text{WT2}}$$

$$\beta'_{\text{data}} = \beta_{\text{data}} + \beta_{\text{data,PTB}} x_{\text{PTB}} + \beta_{\text{data,WT2}} x_{\text{WT2}}$$

$$\alpha^*_{\text{param}} = \alpha_{\text{param,NT}}(1 - x_T) + \alpha_{\text{param,T}} x_T$$

$$\beta^*_{\text{data}} = \beta_{\text{data,NT}}(1 - x_T) + \beta_{\text{data,T}} x_T$$

$$\alpha^\dagger_{\text{param}} = \alpha_{\text{param}} + \alpha_{\text{rate}} \log Y$$

$$\beta^\dagger_{\text{data}} = \beta_{\text{data}} + \beta_{\text{rate}} \log Y$$

Here $x_{\text{PTB}}$, $x_{\text{WT2}}$, and $x_T$ are dummy variables for PTB, WT2, and the transformer respectively. $Y$ is the year, and $\alpha_{\text{rate}}$ and $\beta_{\text{rate}}$ are constants that determine how quickly scaling exponents ($\alpha_{\text{param}}$ and $\beta_{\text{data}}$) change over time.

Models 1 to 11 are all constructed using a similar set of rules:

- $\alpha_{\text{const}}$ and $\beta_{\text{const}}$ determine the coefficients of the parameter and data reducible loss terms respectively

- $\alpha_{\text{year}}$ and $\beta_{\text{year}}$ capture algorithmic progress in parameters and data

- $\alpha_{\text{param}}$ and $\beta_{\text{data}}$ determine the scaling behavior with respect to $N$ and $D$ respectively

All of these parameters may be set as benchmark-specific (e.g. by writing $\alpha'_{\text{const}}$ in lieu of $\alpha_{\text{const}}$).

The remaining models are defined using a different set of rules:

- Model 12 is defined in similar fashion but in 'Hicks-neutral' fashion, such that the same degree of efficiency gain is seen across both parameters and data.

- Model 13 models different scaling exponents $\alpha_{\text{param}}$ and $\beta_{\text{data}}$ for transformer vs non-transformer models

- Model 14 captures only algorithmic progress via changes in the scaling exponents $\alpha^\dagger_{\text{param}}$ and $\beta^\dagger_{\text{data}}$

- Model 15 captures algorithmic progress via $\alpha_{\text{year}}$ and $\beta_{\text{year}}$, as well as changes in the scaling exponents

Table 11: Model specifications for leave-one-out cross validation (Models 1 to 11).

| No. | Model specification |
|---|---|
| 1 | Algorithmic progress in both $N$ and $D$ 
 $L = \exp[\alpha_{\text{const}} - \alpha_{\text{year}}(Y - Y_0) - \alpha_{\text{param}}(\log N - \log N_0)] + \exp[\beta_{\text{const}} - \beta_{\text{year}}(Y - Y_0) - \beta_{\text{data}}(\log D - \log D_0)]$ |
| 2 | No algorithmic progress in parameters $N$ 
 $L = \exp[\alpha_{\text{const}} - \alpha_{\text{param}}(\log N - \log N_0)] + \exp[\beta_{\text{const}} - \beta_{\text{year}}(Y - Y_0) - \beta_{\text{data}}(\log D - \log D_0)]$ |
| 3 | No algorithmic progress in data $D$ 
 $L = \exp[\alpha_{\text{const}} - \alpha_{\text{year}}(Y - Y_0) - \alpha_{\text{param}}(\log N - \log N_0)] + \exp[\beta_{\text{const}} - \beta_{\text{data}}(\log D - \log D_0)]$ |
| 4 | Benchmark-specific in $\alpha_{\text{year}}$ 
 $L = \exp[\alpha_{\text{const}} - \alpha'_{\text{year}}(Y - Y_0) - \alpha_{\text{param}}(\log N - \log N_0)] + \exp[\beta_{\text{const}} - \beta_{\text{year}}(Y - Y_0) - \beta_{\text{data}}(\log D - \log D_0)]$ |
| 5 | Benchmark-specific in $\beta_{\text{year}}$ 
 $L = \exp[\alpha_{\text{const}} - \alpha_{\text{year}}(Y - Y_0) - \alpha_{\text{param}}(\log N - \log N_0)] + \exp[\beta_{\text{const}} - \beta'_{\text{year}}(Y - Y_0) - \beta_{\text{data}}(\log D - \log D_0)]$ |
| 6 | Benchmark-specific in $\alpha_{\text{year}}$ and $\beta_{\text{year}}$ 
 $L = \exp[\alpha_{\text{const}} - \alpha'_{\text{year}}(Y - Y_0) - \alpha_{\text{param}}(\log N - \log N_0)] + \exp[\beta_{\text{const}} - \beta'_{\text{year}}(Y - Y_0) - \beta_{\text{data}}(\log D - \log D_0)]$ |
| 7 | Benchmark-specific in $\alpha_{\text{const}}$ and $\beta_{\text{const}}$ 
 $L = \exp[\alpha'_{\text{const}} - \alpha_{\text{year}}(Y - Y_0) - \alpha_{\text{param}}(\log N - \log N_0)] + \exp[\beta'_{\text{const}} - \beta_{\text{year}}(Y - Y_0) - \beta_{\text{data}}(\log D - \log D_0)]$ |
| 8 | Benchmark-specific in $\alpha_{\text{const}}$ and $\beta_{\text{const}}$, with no algorithmic progress in $N$ 
 $L = \exp[\alpha'_{\text{const}} - \alpha_{\text{param}}(\log N - \log N_0)] + \exp[\beta'_{\text{const}} - \beta_{\text{year}}(Y - Y_0) - \beta_{\text{data}}(\log D - \log D_0)]$ |
| 9 | Benchmark-specific in $\alpha_{\text{const}}$ and $\beta_{\text{const}}$, with no algorithmic progress in $D$ 
 $L = \exp[\alpha'_{\text{const}} - \alpha_{\text{year}}(Y - Y_0) - \alpha_{\text{param}}(\log N - \log N_0)] + \exp[\beta'_{\text{const}} - \beta_{\text{data}}(\log D - \log D_0)]$ |
| 10 | Benchmark-specific in $\alpha_{\text{const}}, \alpha_{\text{year}}, \beta_{\text{const}}, \beta_{\text{year}}$ 
 $L = \exp[\alpha'_{\text{const}} - \alpha'_{\text{year}}(Y - Y_0) - \alpha_{\text{param}}(\log N - \log N_0)] + \exp[\beta'_{\text{const}} - \beta'_{\text{year}}(Y - Y_0) - \beta_{\text{data}}(\log D - \log D_0)]$ |
| 11 | Benchmark-specific in $\alpha_{\text{const}}, \alpha_{\text{year}}, \alpha_{\text{param}}, \beta_{\text{const}}, \beta_{\text{year}}$ and $\beta_{\text{data}}$ 
 $L = \exp[\alpha'_{\text{const}} - \alpha'_{\text{year}}(Y - Y_0) - \alpha'_{\text{param}}(\log N - \log N_0)] + \exp[\beta'_{\text{const}} - \beta'_{\text{year}}(Y - Y_0) - \beta'_{\text{data}}(\log D - \log D_0)]$ |

- Models 16 and 17 do not capture algorithmic progress, and are used as a baseline for goodness-of-fit comparisons. Model 16 is modeled off equation 34, and model 17 modifies this to include transformer-specific scaling exponents. This is described in more detail in Appendix J.2

- Model 18 is a model that only considers the total compute, via the approximation $C = 6ND$. This is discussed in more detail in Appendix J.1.

- Model 19 includes a term that accounts for different vocabulary sizes. We discuss this in Appendix E.2.2.

- Model 20 is the same as model 7, but rather than using purely the training dataset size, it defines "training data" in a way that accounts for the number of epochs of training. Where the number of training epochs is unknown, we impute an epoch count of 1. This model is presented in more detail in Appendix F.

Table 12: Model specifications for leave-one-out cross validation (Models 12 to 20).

| No. | Model specification |
|---|---|
| 12 | 'Hicks-neutral' model
$L = (\exp[\alpha'_{\text{const}} + \alpha_{\text{param}}(\log N - \log N_0)] + \exp[\beta'_{\text{const}} + \beta_{\text{data}}(\log D - \log D_0)]) \cdot \exp[-\alpha_{\text{year}}(Y - Y_0)]$ |
| 13 | Transformer vs non-transformer scaling
$L = \exp[\alpha'_{\text{const}} - \alpha_{\text{year}}(Y - Y_0) - \alpha^*_{\text{param}}(\log N - \log N_0)] + \exp[\beta'_{\text{const}} - \beta_{\text{year}}(Y - Y_0) - \beta^*_{\text{data}}(\log D - \log D_0)]$ |
| 14 | Progress only via changing scaling exponents
$L = \exp[\alpha'_{\text{const}} - \alpha^\dagger_{\text{param}}(\log N - \log N_0)] + \exp[\beta'_{\text{const}} - \beta^\dagger_{\text{data}}(\log D - \log D_0)]$ |
| 15 | Changing $\alpha_{\text{year}}, \beta_{\text{year}}, \alpha_{\text{param}}$, and $\beta_{\text{data}}$
$L = \exp[\alpha'_{\text{const}} - \alpha_{\text{year}}(Y - Y_0) - \alpha^\dagger_{\text{param}}(\log N - \log N_0)] + \exp[\beta'_{\text{const}} - \beta_{\text{year}}(Y - Y_0) - \beta^\dagger_{\text{data}}(\log D - \log D_0)]$ |
| 16 | Benchmark-specific Hoffmann et al. [2022] scaling law (for comparison only)
$L = \exp[\alpha'_{\text{const}} - \alpha_{\text{param}}(\log N - \log N_0)] + \exp[\beta'_{\text{const}} - \beta_{\text{data}}(\log D - \log D_0)]$ |
| 17 | Transformer-specific scaling law (for comparison only)
$L = \exp[\alpha'_{\text{const}} - \alpha^*_{\text{param}}(\log N - \log N_0)] + \exp[\beta'_{\text{const}} - \beta^*_{\text{data}}(\log D - \log D_0)]$ |
| 18 | Compute-only model (for comparison only)
$L = \exp[\alpha'_{\text{const}} - \alpha_{\text{year}}(Y - Y_0) - \alpha_{\text{compute}}(\log(6ND) - \log(6N_0D_0)]$ |
| 19 | Vocabulary fixed-effects (for comparison only)
$L = \gamma \log(\text{vocab}) + \exp[\alpha'_{\text{const}} - \alpha_{\text{year}}(Y - Y_0) - \alpha_{\text{param}}(\log N - \log N_0)] + \exp[\beta'_{\text{const}} - \beta_{\text{year}}(Y - Y_0) - \beta_{\text{data}}(\log D - \log D_0)]$ |
| 20 | Same as model 7 but with imputed epochs (for comparison only)
$L = \exp[\alpha'_{\text{const}} - \alpha_{\text{year}}(Y - Y_0) - \alpha_{\text{param}}(\log N - \log N_0)] + \exp[\beta'_{\text{const}} - \beta_{\text{year}}(Y - Y_0) - \beta_{\text{data}}(\log D - \log D_0)]$ |

We chose to include models 16 to 20 in the cross validation analysis to help the robustness checks performed in different appendices.

## J.1 Compute-only model

Given that our core focus is on estimating doubling times in effective compute, one natural parameterization is to directly consider total training compute $C$, in particular,

$$L = \gamma' + \exp[\alpha'_{\text{const}} - \alpha_{\text{year}}(Y - Y_0) - \alpha_{\text{compute}}(\log C - \log C_0)]. \tag{33}$$

Here $\gamma'$ is defined similarly to the "primed" constants above, i.e. $\gamma' = \gamma + \gamma_{\text{PTB}} x_{\text{PTB}} + \gamma_{\text{WT2}} x_{\text{WT2}}$, and we have the approximate relation that $C \approx 6ND$ [Hoffmann et al., 2022]. However, this model has a tendency to yield implausible results, in particular with a very small scale exponent $\alpha_{\text{compute}}$ and a very short effective compute doubling time (on the order of 1-5 months).

We omit this model because we believe there are two reasons to expect this model to be strongly misspecified. The first reason is that the model scaling does not accurately reflect complementarities between scaling parameter and data. For illustration, compare the following two equations:

$$L = E + \frac{A}{N^\alpha} + \frac{B}{D^\beta} \tag{34}$$

$$L = E + \frac{A}{C^\alpha} = E + \frac{A}{(6ND)^\alpha} \tag{35}$$

where $E$ is the irreducible loss, and $A, B, \alpha, \beta$ are constants. In equation 34, reductions to the loss are bottlenecked by a lack of sufficient data $D$. This bottleneck does not exist in the case of 35, where scaling $N$ arbitrarily would bring you to the irreducible loss $E$. This model thus unrealistically suggests equivalent performance between a 1-parameter model trained on $10^{24}$ tokens, and a $10^{12}$

parameter model trained on $10^{12}$ tokens. Of course, such extreme choices of parameters $N$ and dataset size $D$ are implausible in practice, but nevertheless we do see clear variation in how these values are chosen in practice. For instance, Figure 10 shows a plot $\log_{10}(N/D)$ for models across different years, spanning over five orders of magnitude over this time period. This is in spite of the estimate by Hoffmann et al. [2022] that $N/D$ should be $O(1)$ for compute-optimal training.

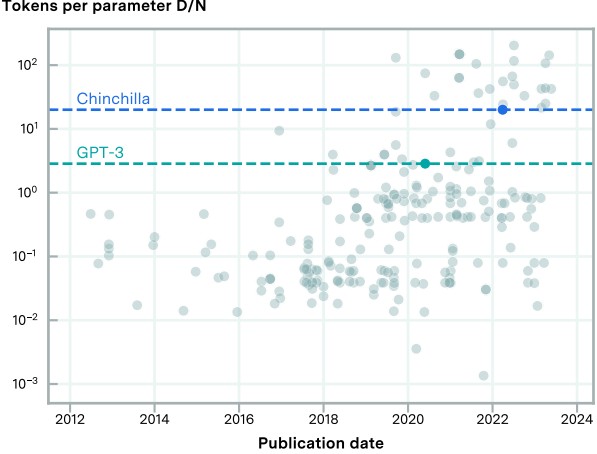

Figure 10: Ratio of parameters $N$ to dataset size $D$ for models in our dataset. We emphasize the ratios corresponding to GPT-3 [Brown et al., 2020] and Chinchilla models [Hoffmann et al., 2022], which are of historical importance in determining how to choose this ratio when scaling language models.

A second way in which equation 33 is misspecified is due to its unrealistic scaling behavior. To see why this is the case, we compare the difference between the reducible loss terms in equations 34 and 35, by writing them as a single fraction with the same denominator $(ND)^{\alpha}$. For illustration purposes, we also assume that $\alpha = \beta$ and $A = B$, which simplify our argument without changing the core conclusion. If we then factor out the coefficient $A$, we then have that

$$L_H \propto \frac{1}{N^{\alpha}} + \frac{1}{D^{\alpha}} = \frac{N^{\alpha} + D^{\alpha}}{(ND)^{\alpha}} \tag{36}$$

$$L_C \propto \frac{1}{(6ND)^{\alpha}} = \frac{6^{-\alpha}}{(ND)^{\alpha}}, \tag{37}$$

where $L_H$ is the loss predicted from the scaling law in Hoffmann et al. [2022], and $L_C$ is the loss from only considering compute. Here we observe that in the case of Chinchilla-scaling (equation 36), as $N$ or $D$ is increased the value of the numerator *increases*, whereas the opposite is true in the compute-only case (equation 37). As a result, fits of the compute-only model tend to have very small values of $\alpha$, since the difference in scaling behaviour between the two expressions tends to be more pronounced for larger values of $\alpha$. Indeed, estimates using the compute-only model yield very small values of $\alpha$, at 0.004 with a 95% CI of [0.002, 0.012]. We illustrate the difference between the scaling behavior of equations 36 and 37 in Figure 11.

We nevertheless test the compute-only model in cross validation and find that it performs very poorly on out-of-sample prediction, far worse than any of the other models that we consider. The results of this exercise are elaborated upon in section J.3.

## J.2   Algorithmic progress through changes in the scale exponents $\alpha_{\text{param}}$ and $\beta_{\text{data}}$

As mentioned in Section 2.1, the final model that we use for our core results does not explicitly account for efficiency improvements through changes in scaling exponents (i.e. $\alpha_{\text{param}}$ and $\beta_{\text{data}}$). The primary reason for our decision is that while this form of algorithmic progress is theoretically plausible, our model fits to the available data suggest that this effect has not been very large. For example, if we consider the estimates from model 15, which includes both the form of algorithmic progress described in Section 2.1 and improvements through changes in scaling exponents, we find that the overall contribution to algorithmic progress is dominated by the former.

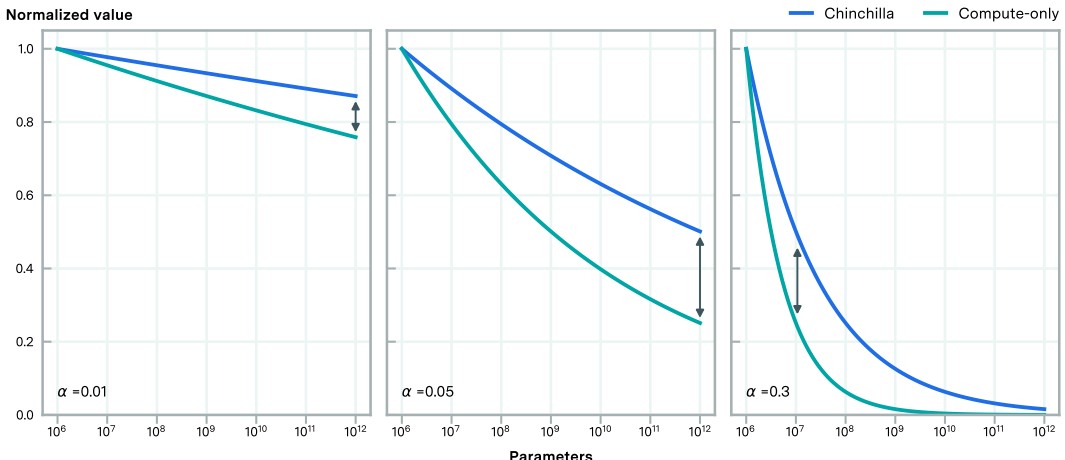

Figure 11: Comparing the scaling behavior of equations 36 and 37, for $\alpha = 0.01$, 0.05 and 0.3. For illustration, we choose $N = D$, and normalize both expressions to equal 1 when the number of parameters $N = 10^6$. The difference between the scaling laws is smallest when $\alpha$ is set to a smaller value—this is illustrated by the double-headed arrow in each plot, showing the largest gap between the two curves.

Another piece of supporting evidence is that in cross validation, model 14 appears to perform roughly as well as models without any algorithmic progress at all (models 16 and 17). Furthermore, the parameters which determine the rate of changing scale exponents ($\alpha_{\text{rate}}$ and $\beta_{\text{rate}}$) are generally very small (around 0.001 to 0.01). As such, the model appears to simply be approximating the equivalent models without algorithmic progress. This suggests that this form of algorithmic progress is negligible, and is also why we have excluded model 14 from Figure 1.

## J.3 Performance metrics

In this section we list the resulting goodness-of-fit metrics from our cross validation analysis. In this case we report the average out-of-sample MSE loss.

| Model/$\delta$-value | 0 | 0.001 | 0.0025 | 0.005 | 0.01 | 0.02 |
|---|---|---|---|---|---|---|
| 1 | 0.04793 | 0.04778 | 0.04762 | 0.04762 | 0.04788 | 0.04997 |
| 2 | 0.05112 | 0.05111 | 0.05049 | 0.05293 | 0.05745 | 0.05793 |
| 3 | 0.04727 | 0.04732 | 0.04758 | 0.04797 | 0.048 | 0.04826 |
| 4 | 0.04716 | 0.04665 | 0.04618 | 0.04656 | 0.04696 | 0.04693 |
| 5 | 0.04457 | 0.04458 | 0.04502 | 0.04545 | 0.04606 | 0.04709 |
| 6 | 0.04539 | 0.04561 | 0.04529 | 0.04519 | 0.04669 | 0.04756 |
| 7 | 0.04795 | 0.04809 | 0.04836 | 0.04676 | 0.0468 | 0.04696 |
| 8 | 0.04671 | 0.04771 | 0.04666 | 0.04689 | 0.04745 | 0.0501 |
| 9 | 0.04622 | 0.04705 | 0.04732 | 0.0461 | 0.04686 | 0.04694 |
| 10 | 0.04405 | 0.04285 | 0.04217 | 0.04642 | 0.04758 | 0.04804 |
| 11 | 0.14378 | 0.04562 | 0.04402 | 0.04504 | 0.04518 | 0.04613 |
| 12 | 0.04845 | 0.04669 | 0.04731 | 0.04615 | 0.04656 | 0.04567 |
| 13 | 0.04381 | 0.04515 | 0.04525 | 0.04454 | 0.04468 | 0.04477 |
| 14 | 0.05144 | 0.06391 | 0.0637 | 0.06298 | 0.06361 | 0.06424 |
| 15 | 0.04751 | 0.04884 | 0.04843 | 0.04611 | 0.04612 | 0.0476 |
| 16 | 0.06448 | 0.0645 | 0.06401 | 0.0629 | 0.0632 | 0.06307 |
| 17 | 0.05668 | 0.05846 | 0.05879 | 0.05821 | 0.05836 | 0.05842 |
| 18 | 0.05542 | 0.05541 | 0.05543 | 0.05545 | 0.05551 | 0.05568 |
| 19 | 0.04643 | 0.04731 | 0.04632 | 0.0467 | 0.04669 | 0.04713 |
| 20 | 0.04564 | 0.04659 | 0.04929 | 0.04751 | 0.04765 | 0.04827 |

Table 13: Average mean squared error test loss of all model-$\delta$ combinations from cross validation. $\delta$-values here are the regularization term in the $L1$ regularization set-up.

