# OpenReview forum: "Algorithmic progress in language models"
_NeurIPS.cc/2024/Conference — NeurIPS 2024 poster_

### Official Review · Reviewer_CCHC · 2024-07-11

**Soundness:** 3
**Presentation:** 3
**Contribution:** 2
**Rating:** 6
**Confidence:** 3

**Summary:**

The authors of this paper examine the performance improvements of language models over the past decade, and investigate how much of it can be attributed to algorithmic improvements of language models.

**Strengths:**

I note here that, given that this paper proposes a method to evaluate the historical progress of language models, I believe that the primary goal for the paper would be to provide interesting insights for the field and outline current open questions. As such, I will structure the rest of my review with that in mind.

- The topic of language models is more central than ever to the broad NeurIPS community, and I think that analysis on the historical progress of the field is of great interest.

- The authors obtain valuable insight on the cause of language model improvements over the years. I find their conclusion that data/compute scaling has been a major driving force for improvement over the past  few years interesting, if somewhat expected.

- The result on the importance of the transformer is also interesting, and provides a nice retrospective justification of the broad adoption of this architecture.

- The analysis of data from past works is also sound.

**Weaknesses:**

- The most crucial weakness of this paper is that, despite the analysis of past trends, it does not provide clear insights or suggestions on where the field should direct its efforts, moving on. The authors acknowledge this limitation, but it seems to me that such discussion Is crucial, in order for this paper to be of interest to the community.

- It is also not clear to me how the insights provided by the paper could extrapolate in the future. While there has been a lot of effort and improvements gained in performance from data scaling, this is not something that can reliably keep on - at some point, sources of data and compute limitations catch up. As such, it is not clear to me how the insight that the effective compute for language models doubles every set period of time can be useful for many years down the line.

- On more minor note, I think some points can be improved for clarity of presentation:

  - The captions of Figures 1a and 1b are joined together - some spacing between them is required.

  -  It would be better if the authors clarified the statistical models used in the main paper, rather than in the appendix.


Overall, my key concern for the paper is that it does not provide enough insight for future directions.

**Questions:**

I would be grateful if the authors could expand on the points I raised above, regarding the insight for future directions.

**Limitations:**

The authors have adequately addressed the limitations of their work. Regarding negative societal impact, I do not foresee any arising from this work.

---

> ### Author Rebuttal · Authors · 2024-08-06
>
> Thank you for the feedback – we address your questions and concerns below.
>
> We fully agree with you that there is substantial uncertainty about how our results could extrapolate into the future. Indeed, we have mentioned this point in the limitations section, and pointed out that our core focus has always been to estimate historical rather than future rates of progress.
>
> That said, although future extrapolations are not within the primary scope of our paper, our results still represent a substantial step up compared to the prior status quo. For one, it provides a rate of progress that can be extrapolated in the first place, and it paves a clear path forward for future work. E.g. We agree with your point that compute limitations might be relevant for understanding future algorithmic progress, and this provides a clear next step of trying to quantify the significance and timing of this bottleneck.
>
> In fact, we feel that this strongly relates to your point about providing insight for future directions. Our work is most strongly directed towards people interested in understanding trends and drivers of progress in ML. Thus while it is of general interest for ML practitioners, by far the most important future directions pertain to future research on ML trends, such as in understanding future compute/data bottlenecks. We also mention that further research could extend the analysis to specific benchmarks or different domains, or consider the impact of individual algorithmic innovations by performing ML experiments. As we alluded to in the related work section, there has been relatively little work studying these important high-level questions about progress in ML and we believe our work points strongly towards additional work in this area.
>
> As such, we completely agree with you that the primary goal of the paper should be to provide interesting insights and outline future directions. We believe that our work achieves these two criteria most directly for people interested in studying ML trends/progress, as per the previous discussion.
>
> As a final point, thanks for the suggestions regarding improving the paper’s presentation – we will incorporate these changes when we next update the paper.

---

> > ### Comment · Reviewer_CCHC · 2024-08-11
> > **Thank you for your comments.**
> >
> > I would like to thank you for your response to my review. I understand better now how the findings of the paper can relate to future work. I believe that further highlighting the above points in the main paper will increase the paper's impact.
> >
> > Given that this was my main concern, I am raising my score.

---

### Official Review · Reviewer_osWG · 2024-07-13

**Soundness:** 2
**Presentation:** 3
**Contribution:** 2
**Rating:** 4
**Confidence:** 4

**Summary:**

This paper investigates the rate of algorithmic progress on the task of language modeling, using a dataset of over 200 LM evaluations on WikiText and Penn Treebank between 2012-2023. The authors fit an augmented scaling law to the data and show that the models are requiring 2x less compute roughly every eight months -- a rate which supersedes Moore's law. Further, the authors find that more recent performance gains have been primarily due to compute scaling and that the contribution of the transformer architecture is roughly equivalent to two years of algorithmic progress in the field.

**Strengths:**

1. This paper presents an interesting take on quantifying the progress on the task of language modeling using an analysis of data collected from over 200 model evaluations in the past 10-11 years.
2. The authors clearly specific the assumptions made in conducting the analysis, which makes the approach quite readable.

**Weaknesses:**

1. There a number of assumptions made to elicit a quantification of algorithmic progress, and the nature of the task itself necessitates those assumptions be made. However, this also means that the resulting analysis is brittle and as such the technical contributions of the work don't rise to the level of a solid contribution.

**Questions:**

1. Can you give examples of the ways in which the core results would change with different scaling law formulations are used?

**Limitations:**

The limitations were adequately addressed.

---

> ### Author Rebuttal · Authors · 2024-08-06
>
> Thank you for the feedback on our paper.
>
> We agree that we make several assumptions in our work quantifying algorithmic progress, and that this introduces uncertainty into our conclusions. However, we do not believe that these assumptions undermine the core results of our paper. We have performed extensive robustness checks for this purpose, e.g. we consider different ways of managing autocorrelation in Appendix I, different ways of estimating doubling times in Appendix G, etc. In each case, our robustness checks provide grounding for our core empirical results. If you do not believe that these robustness checks address your concern, could you please specify why not, and what assumptions you believe need to be addressed?
>
> To address your question about scaling laws in particular, our work considers a range of different model specifications, as outlined in Tables 10 and 11. These models are varied across different degrees of freedom, such as algorithmic progress impacting scaling exponents, progress that is specific to each benchmark, etc. We illustrate the variation across these degrees of freedom in figure 1b.
>
> Furthermore, in appendix C.1, we compare our model parameter estimates to existing work on scaling laws (e.g. Hoffmann et al 2022 and Kaplan et al 2020). We find that our estimated scaling exponents are consistent with theirs within the time range over which the majority of our data lies. In appendix H, we analyze the impact of including an irreducible loss term in the scaling law formulation, and compare this with our core model. Again, our findings are consistent with our core results. Overall we believe that having consistent results across all these changes provides strong evidence in favor of our core results.

---

### Official Review · Reviewer_Va8z · 2024-07-15

**Soundness:** 4
**Presentation:** 4
**Contribution:** 3
**Rating:** 7
**Confidence:** 4

**Summary:**

This paper presents an analysis of the relative contribution of algorithmic progress to overall LM performance gains over the window of 2012-2023. The authors evaluate a large number of potential equation variants for modeling algorithmic progress using leave-one-out CV. By making use of defined notions of effective data/parameters/computer, the authors estimate that 2x less compute is needed to acheive the same performance roughly every 8 months. The authors find that roughly 2/3 of the scaling progress is due to the scaling of physical compute resources, with the remainder being attributed to algorithmic progress. The singular contribution of the Transformer architectures is individually analyzed. Thorough analysis of the techniques applied and their limitations are presented.

**Strengths:**

The paper is clearly written and well-presented. The depth and quality of the anlaysis are exemplary. The methodology of analysis is not highly original, but its application to algorithmic progress is a novel and useful contribution to the community. The limitations of this kind of analysis are well-discussed, which is a useful contribution in its own right.

**Weaknesses:**

Nits:
Figure 2: label the y axes, consider making especially the right plot a bit larger/more readable, the text is very small
Fig 1: fix the spacing between the (a) and (b) captions, they are nearly overlapping

**Questions:**

None.

**Limitations:**

The limitations are thoroughly discussed in the paper. No relevant missing potential negative social impacts.

---

### Official Review · Reviewer_gaW3 · 2024-07-19

**Soundness:** 4
**Presentation:** 2
**Contribution:** 3
**Rating:** 6
**Confidence:** 3

**Summary:**

This paper breaks down the driving force behind language models into two factors: scaling up the amount of compute used to train the models and algorithmic innovations. A statistical model, similar to the scaling law, is built and fitted to analyze the contributions of these two factors. The paper claims that models with the same performance level require half the compute every eight months, reflecting algorithmic improvement. The authors also find that the majority of scaling progress is due to the scaling of physical compute resources rather than algorithmic progress.

**Strengths:**

The paper presents a very interesting and innovative approach to quantifying the algorithmic progress of language models. By covering over 200 language models from the past decade, the empirical foundation for the conclusions drawn is solid. Additionally, the author has performed extensive robustness checks to ensure the validity of their core results.

**Weaknesses:**

While the paper makes several assumptions to quantify algorithmic progress (including the extra coefficients in the augmented scaling law), these assumptions and the many degrees of freedom undermine the robustness of the proposed model and the conclusions drawn. I have doubts about whether the statistical model built can inform future progress in language modeling.

Minor issue: The caption of Figure 1 seems to be incorrect. It should address Figures 1a and 1b instead of 3a and 3b.

**Questions:**

None

**Limitations:**

I do not see any potential negative social impact of this work.

---

> ### Comment · Area_Chair_bt86 · 2024-08-10
> **Is there a mix up?**
>
> Dear reviewer,
>
> is this the review for the write paper? The authors and the area chair suspect that this review is for a different paper. Could you kindly update your review?
>
> Thanks,
> Your Area Chair

---

> > ### Comment · Reviewer_gaW3 · 2024-08-11
> > **Appologise for mix up**
> >
> > Yes, I mistakenly pasted the wrong review. I sincerely apologize to the authors and the area chair for the confusion and inconvenience caused. I have updated my review.

---

### Decision · Program_Chairs · 2024-09-25

**Decision:**

Accept (poster)

**Comment:**

The authors of this paper try to quantify the rate of progress in deep learning over the course of the last decade and disentangle progress from scaling the amount of compute and progress due to algorithmic advances. The research question the authors are trying to tackle is an interesting one and the paper does an adequate job shedding some light on it. There are quite a few assumptions in the work and the estimates might be a bit off, but overall the reviewers found the paper worthy of acceptance and the AC concurs.